# GENERALIZATION GUARANTEES FOR REPRESENTATION LEARNING VIA DATA-DEPENDENT GAUSSIAN MIXTURE PRIORS

**Milad Sefidgaran**[†], **Abdellatif Zaidi**[††], **Piotr Krasnowski**[†]

[†] Paris Research Center, Huawei Technologies France     [†] Université Gustave Eiffel, France

{milad.sefidgaran2,piotr.g.krasnowski}@huawei.com,
 abdellatif.zaidi@univ-eiffel.fr

## ABSTRACT

We establish in-expectation and tail bounds on the generalization error of representation learning type algorithms. The bounds are in terms of the relative entropy between the distribution of the representations extracted from the training and "test" datasets and a data-dependent symmetric prior, i.e., the Minimum Description Length (MDL) of the latent variables for the training and test datasets. Our bounds are shown to reflect the "structure" and "simplicity" of the encoder and significantly improve upon the few existing ones for the studied model. We then use our in-expectation bound to devise a suitable data-dependent regularizer; and we investigate thoroughly the important question of the selection of the prior. We propose a systematic approach to simultaneously learning a data-dependent Gaussian mixture prior and using it as a regularizer. Interestingly, we show that a *weighted attention mechanism* emerges naturally in this procedure. Our experiments show that our approach outperforms the now popular Variational Information Bottleneck (VIB) method as well as the recent Category-Dependent VIB (CDVIB).

## 1 INTRODUCTION

One major problem in learning theory pertains to how to guarantee that a statistical learning algorithm performs on new, unseen data as well as on the used training data, i.e., it has good *generalization* properties. This key question, which has roots in various scientific disciplines, has been studied using seemingly unrelated approaches, including compression-based (LW86; BEHW87; AGNZ18; BL03; SAM+20; HJTW21; BSE+21; HK19; HKS19; BHMZ20; HK21; HKSW20; CK22; SGRS22; SZ24), information-theoretic (RZ16; XR17; SZ20; EGI20; BZV20; HDMR21; NDHR21; ABT+21; HRVSG21; ZTL22; LN22; HD22), PAC-Bayes (See02; LC01; Cat03; Mau04; GLLM09; TS13; BGLR16; TIWS17; DR17; NBS18; RKSST20; NDR20; NHD+20; VGHM21), and intrinsic dimension-based (ŞSDE20; BLGŞ21; HSKM22; LWŞ22) approaches.

In practice, a common approach advocates the usage of a two-part, or *encoder-decoder*, model, often referred to as *representation learning*. In this approach, the encoder part of the model shoots for extracting a "minimal" *representation* of the input (i.e., small generalization error), whereas the decoder part shoots for small empirical risk. One popular approach is the information bottleneck (IB), which was first introduced in (TPB00) and then extended in various ways (SST10; AFDM17; EAZ18; KTW19; Fis20; RGTS20; KASK22). The IB principle is mainly based on the assumption that Shannon's mutual information (MI) between the input and the representation is a good indicator of the generalization error. However, this assumed relationship has been refuted in several works (KTVK18; RG19; AG19; GK19; DKSV20; LLS+23; SZK23). As shown in these works, the few existing theoretical MI-based generalization bounds (*e.g.,* (VPV18; KDJH23)) become vacuous in most reasonable setups. Also, in practice, no consistent relation between the generalization error and the MI has been observed experimentally so far. Rather, recent works (BL03; GK19; SZK23) have shown that the generalization error of representation learning algorithms is related to the *minimum description length* (MDL) of the latent variable and the so-called *geometric compression*. Geometric compression occurs when latent vectors are designed so as to concentrate around a limited number of representatives which form centroid vectors of associated clusters (AG19; GK19). In such

$$X \rightarrow P_{U|X,W_e} \rightarrow U \rightarrow P_{\hat{Y}|U,W_d} \rightarrow \hat{Y}$$

Figure 1: Studied representation learning setup.

settings, inputs can be mapped to the centroids of the clusters that are closest to their associated latent vectors (i.e., lossy compression); and this yields non-vacuous bounds at the expense of only a small (distortion) penalty. The benefit of this lossy compression approach can be appreciated when opposed to classic MI-based bounds (VPV18; KDJH23) which are known to be vacuous when the latent vectors are deterministic functions of the inputs.

In this work, we study the problem of representation learning depicted in Fig. 1 from a generalization error perspective. Then, we use the obtained generalization bound to design and discuss various choices of generalization-inspired regularizers using data-dependent Gaussian mixture priors. To the best knowledge of the authors, generalization error bounds that account suitably for the encoder-decoder structure of the representation learning problem of Fig. 1 are very scarce; and, in fact, with the exception of (SZK23), no non-vacuous bounds for these settings have been reported so far.

**Contributions:** Our main contributions in this work are summarized as follows.

- We establish in-expectation and tail bounds on the generalization error of the representation learning algorithms. Our bounds are expressed in terms of the relative entropy between the distribution of the representations extracted from training and "test" datasets and a data-dependent symmetric prior $\mathbf{Q}$, i.e., the Minimum Description Length ($\mathrm{MDL}(\mathbf{Q})$) of the latent variables for training and test datasets – (Bounds that depend on $\mathrm{MDL}(\mathbf{Q})$ are arguably *better* bounds because they capture the structure and simplicity of the encoders in sharp contrast with IB-based approaches (BL03)). However, our bounds are shown to be possibly tighter than those of (SZK23). For instance, while the bounds of (SZK23) are of the order of $\sqrt{\mathrm{MDL}(\mathbf{Q})/n}$, where $n$ designates the size of the used training dataset, ours is approximately of the order of $\mathrm{MDL}(\mathbf{Q})/n$ for the realizable setup.

- We propose a systematic approach to finding a suitable "data-dependent" prior that we then use to construct judiciously a regularizer during training (based on our newly established bounds). Specifically, first, we observe that if the latent variables are generated according to a Gaussian distribution, then the prior $\mathbf{Q}$ that minimizes the *empirical* $\mathrm{MDL}(\mathbf{Q})$ term is a Gaussian mixture distribution. Then, using this and the known fact that Gaussian mixture distributions can approximate sufficiently well any arbitrary distribution when the number of mixture components is large enough (DH83; GBC16; NND$^+$22), we propose two methods for simultaneously finding a Gaussian mixture prior and using it as a regularizer along the optimization iterations. The methods are coined 'lossless Gaussian mixture prior" and "lossy Gaussian mixture prior", respectively. In essence, the procedure consists of finding the underlying "structure" of the latent variables in the form of a Gaussian mixture prior; and, simultaneously, steers the latent variables to best fit with this found structure. Interestingly, in the lossy version of the approach, which is shown to generally yield better performance, the components of the Gaussian mixture are updated using a mechanism that is similar to the self-attention mechanism. In particular, the components are updated according to the extent they each "*attend*" to the latent variables statistically.

- We validate the advantages of our generalization-aware regularizer in practice through experiments using various datasets (CIFAR10, CIFAR100, INTEL, and USPS) and encoder architectures (CNN4 and ResNet18). In particular, we show that our approach outperforms the popular VIB of (AFDM17) and the recent Category-Dependent VIB of (SZK23). The reader is referred to Section 5 and Appendix E for details on the datasets, models, and experiments.

We emphasize once more that our approach here, which measures complexity using MDL of the involved latent variables, has two appealing features: (i) it yields generalization bounds that only depend on the encoder part of representation type statistical learning algorithms, and (ii) the employed lossy compression enables the yielded bounds to only take finite values, i.e., not vacuous, in reasonable setups, by opposition to the MI bounds of (VPV18; KDJH23). The described approach and results must be contrasted with classes of prior-art bounds that measure complexity differently. The first class

of bounds involves the complexity of the hypothesis (model) space and includes, e.g., MI-based, PAC-Bayes, and some of the compression-based bounds (e.g. (AGNZ18)). Such bounds mostly involve "data-independent" priors on the model; and seldom use "data-dependent" priors (DR18; PORSTS21) – see (Alq21, Section 3.3) for a detailed review. Generalization bounds that use model complexity do not seem to be amenable to using them for regularization since, in practice, one has only a single instance of the posterior. The second class of bounds are intrinsic dimension-based bounds that measure the complexity of the model along the optimization trajectories. Although in this approach multiple instances of the posterior are available, measuring the trajectory complexity of large models is not practical. The third class of bounds uses prediction complexity such as with f-CMI (HRVSG21; HD22) - see also the related (BL03; SZK23). In such bounds, typically the complexity appears in both the loss function and the regularizer; and this is generally not reasonable in practice.

**Notations.** We denote the random variables and their realizations by upper and lower case letters and use Calligraphy fonts to refer to their support set *e.g.,* $X$, $x$, and $\mathcal{X}$. The distribution of $X$ is denoted by $P_X$,[1] which for simplicity, is assumed to be a *probability mass function* for a random variable with discrete support set and to be *probability density function* otherwise. With this assumption, the Kullback–Leibler (KL) between two distributions $P$ and $Q$ is defined as $D_{KL}(P\|Q) := \mathbb{E}_P[\log(P/Q)]$ if $P \ll Q$, and $\infty$ otherwise. Lastly, we denote the set $\{1,\ldots,n\}$, $n \in \mathbb{N}^*$, by $[n]$.

## 2 PROBLEM SETUP

We consider a $C$-class classification setup, as described below.

**Data.** We assume that the *input data* $Z$, which take value according to an unknown distribution $\mu$, is composed of two parts $Z = (X, Y)$, where **(i)** $X$ represents the *feature* of the input data, taking values in the *feature space* $\mathcal{X}$, and **(ii)** $Y \in \mathcal{Y}$ represents the label ranging from 1 to $C$, *i.e.,* $\mathcal{Y} = [C]$. We denote the underlying distribution of $X$ and $Y$ by $\mu_X$ and $\mu_Y$, respectively, and their joint distribution by $\mu := \mu_{X|Y}\mu_Y := \mu_X\mu_{Y|X}$.

**Training dataset.** To learn a model, we assume the availability of a *training dataset* $S = \{Z_1,\ldots,Z_n\} \sim \mu^{\otimes n} =: P_S$, composed of $n$ i.i.d. samples $Z_i = (X_i, Y_i)$ of the input data. In our analysis, we often use a *test* dataset (known also as *ghost* dataset (SZ20)) $S' = \{Z'_1,\ldots,Z'_n\} \sim \mu^{\otimes n} =: P_{S'}$, where $Z'_i = (X'_i, Y'_i)$. To simplify the notation, we denote the features and labels of $S$ and $S'$ by $\mathbf{X} := X^n \sim \mu_X^{\otimes n}$, $\mathbf{Y} := Y^n \sim \mu_Y^{\otimes n}$, $\mathbf{X}' := X'^n \sim \mu_X^{\otimes n}$, and $\mathbf{Y}' := Y'^n \sim \mu_Y^{\otimes n}$, respectively.

**Encoder-decoder model.** We assume that the model (hypothesis) is composed of two parts: an encoder and a decoder part. The encoder $w_e \in \mathcal{W}_e$ takes as input the feature $x$ and generates as output the *representation* or the *latent variable* $U \in \mathcal{U}$, possibly stochastically. For simplicity, we assume that $\mathcal{U} = \mathbb{R}^d$, for some $d \in \mathbb{N}^*$. The decoder $w_d \in \mathcal{W}_d$ takes $U$ as input and outputs an estimate $\hat{Y}$ of the true label $Y$. The overall model is denoted by $w := (w_e, w_d) \in \mathcal{W} = \mathcal{W}_e \times \mathcal{W}_d$. The setup is shown in Fig. 1.

**Learning algorithm.** We consider a general stochastic learning framework in which the learning algorithm $\mathcal{A}: \mathcal{Z}^n \to \mathcal{W}$ has access to a training dataset $S$ and uses it to choose a model (or hypothesis) $\mathcal{A}(S) = W \in \mathcal{W}$, where $W = (W_e, W_d)$. The distribution induced by the learning algorithm $\mathcal{A}$ is denoted by $P_{W|S} = P_{W_e,W_d|S}$. Also, the joint distribution of $(S, W)$ is denoted by $P_{S,W}$ and the marginal distribution of $W$ under this distribution is denoted by $P_W$. Furthermore, we denote the induced conditional distribution of the latent variable $U$ given the encoder and the input by $P_{U|X,W_e}$. Finally, we denote the conditional distribution of the model's prediction $\hat{Y}$, conditioned on the decoder and the latent variable, by $P_{\hat{Y}|U,W_d}$. It is easy to see that $P_{\hat{Y}|X,W} = \mathbb{E}_{U \sim P_{U|X,W_e}}[P_{\hat{Y}|U,W_d}]$. Lastly and as a general rule, we use the following shorthand notation

$$P_{\mathbf{U},\mathbf{U}'|\mathbf{X},\mathbf{X}',W_e} := \bigotimes_{i \in [n]}\{P_{U_i|X_i,W_e}P_{U'_i|X'_i,W_e}\}. \tag{1}$$

Similar notation is used to shorten products of distributions, *e.g.,* $P_{\mathbf{U}|\mathbf{X},W_e}$ and $P_{\hat{\mathbf{Y}}|\mathbf{X},W}$.

---

[1] We, however, make an exception for the input data, whose distribution is denoted by $\mu$, as it is common in theoretical papers, e.g. (XR17; BZV20; LN22).

**Risks.** The quality of a model $w$ is assessed by the below 0-1 loss function $\ell \colon \mathcal{Z} \times \mathcal{W} \to \{0,1\}$:

$$\ell(z,w) \coloneqq \mathbb{E}_{\hat{Y} \sim P_{\hat{Y}|x,w}} \big[ \mathbb{1}_{\{y \neq \hat{Y}\}} \big] = \mathbb{E}_{U \sim P_{U|x,w_e}} \mathbb{E}_{\hat{Y} \sim P_{\hat{Y}|U,w_d}} \Big[ \mathbb{1}_{\{y \neq \hat{Y}\}} \Big]. \tag{2}$$

In learning theory, the ultimate goal is to find a model that minimizes the *population risk*, defined as $\mathcal{L}(w) \coloneqq \mathbb{E}_{Z \sim \mu}[\ell(Z,w)]$. However, since the underlying distribution $\mu$ is unknown, only the *empirical risk*, defined as $\hat{\mathcal{L}}(s,w) \coloneqq \frac{1}{n} \sum_{i \in [n]} \ell(z_i,w)$, is accessible and can be minimized. Therefore, a central question in learning theory and this paper is to control the difference between these two risks, known as *generalization error*:

$$\mathrm{gen}(s,w) \coloneqq \mathcal{L}(w) - \hat{\mathcal{L}}(s,w). \tag{3}$$

In our results, for simplicity, we also use the following shorthand notations:

$$\hat{\mathcal{L}}(\mathbf{y}, \hat{\mathbf{y}}) \coloneqq \frac{1}{n} \sum_{i \in [n]} \mathbb{1}_{\{\hat{y}_i \neq y_i\}}, \qquad \hat{\mathcal{L}}(\mathbf{y}', \hat{\mathbf{y}}') \coloneqq \frac{1}{n} \sum_{i \in [n]} \mathbb{1}_{\{\hat{y}'_i \neq y'_i\}}, \tag{4}$$

Note that

$$\hat{\mathcal{L}}(s,w) = \mathbb{E}_{\hat{\mathbf{Y}} \sim P_{\hat{\mathbf{Y}}|\mathbf{x},w}} \Big[ \hat{\mathcal{L}}(\mathbf{y}, \hat{\mathbf{Y}}) \Big], \qquad \hat{\mathcal{L}}(s',w) = \mathbb{E}_{\hat{\mathbf{Y}}' \sim P_{\hat{\mathbf{Y}}'|\mathbf{x}',w}} \Big[ \hat{\mathcal{L}}(\mathbf{y}', \hat{\mathbf{Y}}') \Big]. \tag{5}$$

**Symmetric prior.** Our results are stated in terms of the KL-divergence between a posterior (*e.g.*, $P_{\mathbf{U},\mathbf{U}'|\mathbf{X},\mathbf{X}',W_e}$) and a prior $\mathbf{Q}$ that needs to satisfy some symmetry property.

**Definition 1** (Symmetric prior). *A conditional prior* $\mathbf{Q}(U^{2n}|Y^{2n}, X^{2n}, W_e)$ *is said to be symmetric if* $\mathbf{Q}(U_\pi^{2n}|Y^{2n}, X^{2n}, W_e)$ *is invariant under all permutations* $\pi \colon [2n] \mapsto [2n]$ *for which* $\forall i \colon Y_i = Y_{\pi(i)}$.

## 3 GENERALIZATION BOUNDS FOR REPRESENTATION LEARNING ALGORITHMS

In this section, we establish novel in-expectation and tail bounds on the generalization error of representation learning algorithms for the setup of Fig. 1.

### 3.1 IN-EXPECTATION BOUND

Define the function $h_D \colon [0,1] \times [0,1] \to [0,2]$ as

$$h_D(x_1, x_2) \coloneqq 2h_b\Big(\frac{x_1 + x_2}{2}\Big) - h_b(x_1) - h_b(x_2),$$

where $h_b(x) = -x \log_2(x) - (1-x) \log_2(1-x)$ is the binary Shannon entropy function. It is easy to see that $h_D(x_1, x_2)/2$ equals the Jensen-Shannon divergence between two binary Bernoulli distributions with parameters $x_1 \in [0,1]$ and $x_2 \in [0,1]$. Also, let the function $h_C \colon [0,1] \times [0,1] \times \mathbb{R}^+ \to \mathbb{R}^+$ be defined as

$$h_C(x_1, x_2; \epsilon) \coloneqq \max_{\epsilon'} \Big\{ h_b(x_{1 \wedge 2} + \epsilon') - h_b(x_{1 \wedge 2}) + h_b(x_{x_{1 \vee 2}} - \epsilon') - h_b(x_{x_{1 \vee 2}}) \Big\}, \tag{6}$$

where $x_{1 \wedge 2} = \min(x_1, x_2)$, $x_{1 \vee 2} = \max(x_1, x_2)$, and the maximization in (6) is over all

$$\epsilon' \in \Big[0, \min\Big(\epsilon, \frac{x_{1 \vee 2} - x_{1 \wedge 2}}{2}\Big)\Big]. \tag{7}$$

Hereafter we sometimes use the handy notation

$$h_{\mathbf{y},\mathbf{y}',\hat{\mathbf{y}},\hat{\mathbf{y}}'}(\epsilon) \coloneqq h_C\Big(\hat{\mathcal{L}}(\mathbf{y}, \hat{\mathbf{y}}), \hat{\mathcal{L}}(\mathbf{y}', \hat{\mathbf{y}}'); \epsilon\Big). \tag{8}$$

Now, we state our in-expectation generalization bound for representation learning algorithms.

**Theorem 1.** *Consider a $C$-class classification problem and a learning algorithm $\mathcal{A} \colon \mathcal{Z}^n \to \mathcal{W}$ that induces the joint distribution* $(S', S, W, \mathbf{U}, \mathbf{U}', \hat{\mathbf{Y}}, \hat{\mathbf{Y}}') \sim P_{S'} P_{S,W} P_{\mathbf{U},\mathbf{U}'|\mathbf{X},\mathbf{X}',W_e} P_{\hat{\mathbf{Y}},\hat{\mathbf{Y}}'|\mathbf{U},\mathbf{U}',W_d}$.

*Then, for any symmetric conditional distribution* $\mathbf{Q}(\mathbf{U},\mathbf{U}'|\mathbf{Y},\mathbf{Y}',\mathbf{X},\mathbf{X}',W_e)$ *and for* $n \geqslant 10$, *we have*

$$\mathbb{E}_{\mathbf{S},\mathbf{S}',W,\hat{\mathbf{Y}},\hat{\mathbf{Y}}'}\Big[h_D\Big(\hat{\mathcal{L}}(\mathbf{Y}',\hat{\mathbf{Y}}'),\hat{\mathcal{L}}(\mathbf{Y},\hat{\mathbf{Y}})\Big)\Big] \leqslant$$

$$\frac{\mathrm{MDL}(\mathbf{Q}) + \log(n)}{n} + \mathbb{E}_{\mathbf{Y},\mathbf{Y}',\hat{\mathbf{Y}},\hat{\mathbf{Y}}'}\Big[h_{\mathbf{Y},\mathbf{Y}',\hat{\mathbf{Y}},\hat{\mathbf{Y}}'}\Big(\frac{1}{2}\|\hat{p}_{\mathbf{Y}} - \hat{p}_{\mathbf{Y}'}\|_1\Big)\Big], \quad (9)$$

*where* $\hat{p}_{\mathbf{Y}}$ *and* $\hat{p}_{\mathbf{Y}'}$ *are empirical distributions of* $\mathbf{Y}$ *and* $\mathbf{Y}'$, *respectively, and*

$$\mathrm{MDL}(\mathbf{Q}) := \mathbb{E}_{S,S',W_e}\big[D_{KL}\big(P_{\mathbf{U},\mathbf{U}'|\mathbf{X},\mathbf{X}',W_e}\big\|\mathbf{Q}\big)\big]. \tag{10}$$

The proof of Theorem 1, which appears in Appendix G.1, consists of two main proof steps, a change of measure argument followed by the computation of a moment generation function (MGF). Specifically, we use the Donsker-Varadhan's lemma (DV75, Lemma 2.1) to change the distribution of the latent variables from $P_{\mathbf{U},\mathbf{U}'|\mathbf{X},\mathbf{X}',W_e}$ to $\mathbf{Q}$. This change in measure results in a penalty term equal to $\mathrm{MDL}(\mathbf{Q})$. Let $f$ be given by $n$ times the difference of $h_D$ and the term on the right-hand-side (RHS) of (9) , i.e., $f = n(h_D - \mathrm{RHS}(9))$. We apply the Donsker-Varadhan change of measure on the function $f$, in sharp contrast with related proofs in MI-based bounds literature (XR17; SZ20; Alq21). The second step consists of bounding the MGF of $nf$. For every label $c \in [C]$, let $\mathcal{B}_c$ denote the set of those samples of $S$ and $S'$ that have label $c$. By construction, any arbitrary reshuffling of the latent variables associated with the samples in the set $\mathcal{B}_c$ preserves the labels. In addition, such reshuffling does not change the value of the symmetric prior $\mathbf{Q}$. The rest of the proof consists of judiciously bounding the MGF of $nf$ under the uniform distribution induced by such reshuffles.

It is easy to see that the left hand side (LHS) of (9) is related to the expected generalization error. For instance, since by (SZK23, Lemma 1) the function $h_D(x_1, x_2)$ is convex in both arguments, $h_D(x_1, 0) \geqslant x_1$, and $h_D(x_1, x_2) \geqslant (x_1 - x_2)^2$ for $x_1, x_2 \in [0, 1]$, one has that

$$\mathbb{E}_{\mathbf{S},W}\big[\mathrm{gen}(S, W)\big] \leqslant \mathbb{E}_{\mathbf{S},\mathbf{S}',W,\hat{\mathbf{Y}},\hat{\mathbf{Y}}'}\big[h_D\big(\hat{\mathcal{L}}(\mathbf{Y}',\hat{\mathbf{Y}}'),\hat{\mathcal{L}}(\mathbf{Y},\hat{\mathbf{Y}})\big)\big],$$

and

$$\mathbb{E}_{\mathbf{S},W}\big[\mathrm{gen}(S, W)\big]^2 \leqslant \mathbb{E}_{\mathbf{S},\mathbf{S}',W,\hat{\mathbf{Y}},\hat{\mathbf{Y}}'}\big[h_D\big(\hat{\mathcal{L}}(\mathbf{Y}',\hat{\mathbf{Y}}'),\hat{\mathcal{L}}(\mathbf{Y},\hat{\mathbf{Y}})\big)\big],$$

for the "realizable" and "unrealizable" cases, respectively.

Several remarks are now in order. First, note that the generalization gap bound of Theorem 1 does *not* depend on the classification head; it only depends on the encoder part! In particular, this offers a theoretical justification of the intuition that in representation-type neural architectures the main goal of the encoder part is to seek a good generalization capability whereas the main goal of the decoder part is to seek to minimize the empirical risk. Also, it allows the design of regularizers that depend only on the encoder, namely the complexity of the latent variables, as we will elaborate on thoroughly in the next section. (2) The dominant term of the RHS of (9) is $\mathrm{MDL}(\mathbf{Q})/n$. This can be seen by noticing that the total variation term $\|\hat{p}_{\mathbf{Y}} - \hat{p}_{\mathbf{Y}'}\|_1$ is of the order $\sqrt{C/n}$ as shown in (BK12, Theorem 2); and, hence, the residual

$$B_{\mathrm{emp\_diff}} := \mathbb{E}_{\mathbf{Y},\mathbf{Y}',\hat{\mathbf{Y}},\hat{\mathbf{Y}}'}\Big[h_{\mathbf{Y},\mathbf{Y}',\hat{\mathbf{Y}},\hat{\mathbf{Y}}'}\Big(\frac{1}{2}\|\hat{p}_{\mathbf{Y}} - \hat{p}_{\mathbf{Y}'}\|_1\Big)\Big], \tag{11}$$

is small for large $n$ (see below for additional numerical justification of this statement). (3) The term $\mathrm{MDL}(\mathbf{Q})$, as given by (10), expresses the average (w.r.t. data and training stochasticity) of KL-divergence terms of the form $D_{KL}(\mathbf{P}\|\mathbf{Q})$ where $\mathbf{P}$ is the distribution of the representation in the training samples $n$ and the test samples $n$ conditioned on the features of the $2n$ examples for a given encoder, while $\mathbf{Q}$ is a fixed symmetric prior distribution for representations given $2n$ samples for the given encoder. As stated in Definition 1, $\mathbf{Q}$ is symmetric for any permutation $\pi$; and, in a sense, this means that $\mathbf{Q}$ induces a distribution on $(\mathbf{U},\mathbf{U}')$ conditionally given $(\mathbf{Y},\mathbf{Y}',\mathbf{X},\mathbf{X}',W_e)$ that is invariant under all permutations that preserve the labels of training and ghost samples. (4) The minimum description length of the representations arguably reflects the encoder's "structure" and "simplicity" (SZK23). In contrast, mutual information (MI) type bounds and regularizers, used, e.g., in the now popular IB method, are known to fall short of doing so (Gei21; AG19; RG19; DKSV20; LLS$^+$23). In fact, as mentioned in these works, most existing theoretical MI-based generalization bounds (*e.g.,* (VPV18; KDJH23)) become vacuous in reasonable setups. In addition, no consistent

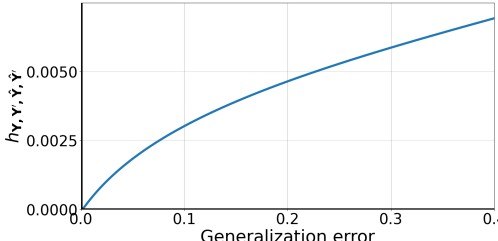

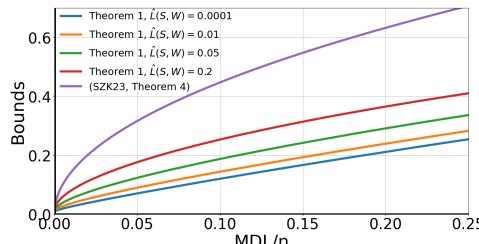

Figure 2: Values of $h_C\left(\hat{\mathcal{L}}(\mathbf{y}, \hat{\mathbf{y}}), \hat{\mathcal{L}}(\mathbf{y}', \hat{\mathbf{y}}'); \epsilon\right)$ for various values of the generalization error for the CIFAR10 dataset.

Figure 3: Comparison of the generalization bounds of Theorem 1 (for various values of $\hat{\mathcal{L}}(S, W)$) and (SZK23, Theorem 4) for the CIFAR10 dataset.

relation between the generalization error and MI has been reported experimentally so far. Therefore, MDL is a better indicator of the generalization error than the mutual information used in the IB principle.

As we already mentioned, the total variation $\|\hat{p}_{\mathbf{Y}} - \hat{p}_{\mathbf{Y}'}\|_1$ is of the order $\sqrt{C/n}$ (BK12, Theorem 2); and for this reason, the second term on the RHS of (9) is negligible in practice. Figure 2 shows the values of the term inside the expectation of $B_{\text{emp\_diff}}$ as given by (11) for the CIFAR10 dataset for various values of the generalization error. The values are obtained for empirical risk of $0.05$ and $\|\hat{p}_{\mathbf{Y}} - \hat{p}_{\mathbf{Y}'}\|_1$ set to be of the order $\sqrt{C/n}$. As it is visible from the figure, the term inside the expectation of $B_{\text{emp\_diff}}$ is the order of magnitude smaller than the generalization error. This illustrates that even for settings with moderate dataset size such as CIFAR, the generalization bound of Theorem 1 is mainly dominated by $\text{MDL}(\mathbf{Q})/n$.

As stated in the Introduction section, generalization bounds for the representation learning setup of Fig. 1 are rather scarce; and, to the best of our knowledge, the only non-vacuous existing in-expectation bound was provided recently in (SZK23, Theorem 4). This bound states that

$$\mathbb{E}_{\mathbf{S}, W}[\text{gen}(S, W)] \leqslant \sqrt{\frac{2\,\text{MDL}(\mathbf{Q}) + C + 2}{n}}, \tag{12}$$

where $C$ is the number of classes.

i. Investigating (9) and (12), it is easy to see that, order-wise, while the bound of (SZK23, Theorem 4) evolves as $\mathcal{O}\left(\sqrt{\text{MDL}(\mathbf{Q})/n}\right)$ our bound of Theorem 1 is tighter comparatively and it evolves approximately as $\mathcal{O}(\text{MDL}(\mathbf{Q})/n)$ for realizable setups with large $n$ (i.e., for most settings in practice).

ii. Figure 3 depicts the evolution of both bounds as a function of $\text{MDL}(\mathbf{Q})/n$ for the CIFAR10 dataset and for different values of the empirical risk. It is important to emphasize that, in doing so, we account for the contribution of all terms of the RHS of (9), including the residual $B_{\text{emp\_diff}}$ which is then *not* neglected. As is clearly visible from the figure, our bound of Theorem 1 is tighter comparatively. Also, the advantage over (12) becomes larger for smaller values of the empirical risk and larger values of $\text{MDL}(\mathbf{Q})/n$.

## 3.2 TAIL BOUND

The following theorem provides a probability tail bound on the generalization error of the representation learning setup of Fig. 1.

**Theorem 2.** *Consider the setup of Theorem 1 and consider some symmetric conditional distribution* $\mathbf{Q}(\mathbf{U}, \mathbf{U}'|\mathbf{Y}, \mathbf{Y}', \mathbf{X}, \mathbf{X}', W_e)$. *Then, for any $\delta \geqslant 0$ and for $n \geqslant 10$, with probability at least $1 - \delta$ over choices of $(S, S', W)$, it holds that*

$$h_D\left(\hat{\mathcal{L}}(S', W), \hat{\mathcal{L}}(S, W)\right) \leqslant \frac{D_{KL}\left(P_{\mathbf{U}, \mathbf{U}'|\mathbf{X}, \mathbf{X}', W_e} \| \mathbf{Q}\right) + \log(n/\delta)}{n}$$

$$+ \mathbb{E}_{\hat{\mathbf{Y}}, \hat{\mathbf{Y}}'|\mathbf{Y}, \mathbf{Y}'} \left[ h_{\mathbf{Y}, \mathbf{Y}', \hat{\mathbf{Y}}, \hat{\mathbf{Y}}'} \left( \frac{1}{2} \|\hat{p}_{\mathbf{Y}} - \hat{p}_{\mathbf{Y}'}\|_1 \right) \right], \qquad (13)$$

where $\hat{p}_{\mathbf{Y}}$ and $\hat{p}_{\mathbf{Y}'}$ are empirical distributions of $\mathbf{Y}$ and $\mathbf{Y}'$, respectively.

The proof of Theorem 2 appears in Appendix G.2.

### 3.3 LOSSY GENERALIZATION BOUNDS

The bounds of the previous section can be regarded as lossless versions of ones that are more general, and which we refer to as *lossy* bounds. The lossy bounds are rather easy extensions of the corresponding lossless versions, but they have the advantage of being guaranteed to stay non-vacuous even when the encoder is set to be deterministic. Also, such bounds are useful to explain the empirically observed *geometrical compression* phenomenon (Gei21). For comparison, MI-based bounds, such as Xu-Raginsky (XR17) are known to suffer both shortcomings (HRGT$^+$23; Liv23). The aforementioned shortcomings have been shown that can be addressed using the lossy approach (SGRS22; SZ24). For the sake of brevity, in the rest of this section we only illustrate how the bound (12) can be extended to a corresponding lossy one. Let $\hat{W}_e \in \mathcal{W}_e$ be any quantized model defined by $P_{\hat{W}_e|S}$, that satisfy the *distortion* criterion $\mathbb{E}_{P_{S,W} P_{\hat{W}_e|S}} \left[ \mathrm{gen}(S, W) - \mathrm{gen}(S, \hat{W}) \right] \leqslant \epsilon$, where $\hat{W} = (\hat{W}_e, W_d)$. Then, we get

$$\mathbb{E}_{\mathbf{S}, W} [\mathrm{gen}(S, W)] \leqslant \sqrt{\frac{2 \, \mathrm{MDL}(\mathbf{Q}) + C + 2}{n}} + \epsilon, \qquad (14)$$

where now $\mathrm{MDL}(\mathbf{Q})$ is considered for the quantized encoder, *i.e.,*

$$\mathrm{MDL}(\mathbf{Q}) \coloneqq \mathbb{E}_{S, S', \hat{W}_e} \left[ D_{KL} \left( P_{\mathbf{U}, \mathbf{U}'|\mathbf{X}, \mathbf{X}', \hat{W}_e} \| \mathbf{Q}(\mathbf{U}, \mathbf{U}'|S, S', \hat{W}_e) \right) \right]. \qquad (15)$$

## 4 REGULARIZATION USING DATA-DEPENDENT GAUSSIAN MIXTURE PRIORS

Theorems 1 and 2 essentially mean that if for a given learning algorithm the minimum description length $\mathrm{MDL}(\mathbf{Q})$ is small, then the algorithm is guaranteed to generalize well. Hence, it is natural to use the term $\mathrm{MDL}(\mathbf{Q})$ as a suitable regularizer. The question of the choice of the prior $\mathbf{Q}$ is pivotal for this. In this section, we propose an effective method to simultaneously find a data-dependent $\mathbf{Q}$ and use it to build a suitable regularizer term along the optimization iterations.

We assume that for a given input $x$ the encoder outputs the mean $\mu_x \in \mathbb{R}^d$ and standard deviation $\sigma_x \in \mathbb{R}^d$. Also, we assume that the latent variable $U$ is distributed according to a multivariate Gaussian distribution with a diagonal covariance matrix, *i.e.,* $U \sim \mathcal{N}(\mu_x, \mathrm{diag}(\sigma_x^2))$ where $\mathrm{diag}(\sigma_x^2)$ denotes a $d \times d$ diagonal matrix with diagonal elements $\sigma_x^2$. With this assumption, we have

$$P_{\mathbf{U}, \mathbf{U}'|\mathbf{X}, \mathbf{X}', W_e} = \bigotimes_{i \in [n]} \left\{ \mathcal{N}(\mu_{x_i}, \mathrm{diag}(\sigma_{x_i}^2)) \mathcal{N}(\mu_{x_i'}, \mathrm{diag}(\sigma_{x_i'}^2)) \right\}.$$

In our approach, we model the prior $\mathbf{Q}$ as a suitable *Gaussian mixture*, with the mixture coefficients chosen judiciously in a manner that is training-data dependent and along the optimization iterations. The rationale for this choice is two-fold: (i) The Gaussian mixture distribution is known to possibly approximate well enough any arbitrary distribution provided that the number of mixture components is sufficiently large (DH83; GBC16) (see also (NNL$^+$22, Theorem 1)); and (ii) given distributions $\{p_i\}_{i \in [N]}$, the distribution $q$ that minimizes $\sum_{i \in [N]} D_{KL}(p_i \| q)$ is $q = \frac{1}{N} \sum_{i \in [N]} p_i$. Thus, if all distributions $p_i$ are Gaussian, the minimizer is a Gaussian mixture.

Let, for $c \in [C]$, $Q_c$ denote the data-dependent Gaussian mixture prior $Q_c$ for label $c$. Also, let $\mathbf{Q}(\mathbf{U}, \mathbf{U}'|S, S', \hat{W}_e) = \prod_{i \in [n]} Q_{Y_i}(U_i) Q_{Y_i'}(U_i')$. It is easy to see that this prior satisfies the symmetry property of Definition 1. In what follows, we explain how the priors $\{Q_c\}$ are chosen and updated along the optimization iterations. As it will become clearer, our method is somewhat reminiscent of the expectation-maximization (EM) algorithm for finding Gaussian mixture priors that maximize the log-likelihood, but with notable major differences: **(i)** In our case the prior must be learned along the optimization iterations with the underlying distribution of the latent variables

possibly changing at every iteration. **(ii)** The Gaussian mixture prior is intended to be used in a regularizer term, not to maximize the log-likelihood; and, hence, the approach must be adapted accordingly. **(iii)** Unlike the usual scenario where the goal is to find an appropriate Gaussian mixture given a set of points, here we are given a set of distributions *i.e.*, $\mathcal{N}\big(\mu_{x_i}, \mathrm{diag}(\sigma_{x_i}^2)\big)$ that generate such points. **(iv)** The found prior must satisfy (at least partially)[2] certain "symmetry" properties.

### 4.1 LOSSLESS GAUSSIAN MIXTURE PRIOR

For each label $c \in [C]$, we let the prior $Q_c$ to be defined as

$$Q_c = \sum\nolimits_{m \in [M]} \alpha_{c,m} \, Q_{c,m}, \tag{16}$$

over $\mathbb{R}^d$, where $\alpha_{c,m} \in [0,1]$, $\sum_{m \in [M]} \alpha_{c,m} = 1$ for each $c \in [C]$, and where $\{Q_{c,m}\}_{c,m}$ are multivariate Gaussian distributions with a diagonal covariance matrix:

$$Q_{c,m} = \mathcal{N}\big(\mu_{c,m}, \mathrm{diag}\big(\sigma_{c,m}^2\big)\big), \quad m \in [M], c \in [C].$$

With the above prior choice, the regularizer term simplifies as $\sum_{i \in [b]} D_{KL}\big(P_{U_i|X_i,W_e} \big\| Q_{Y_i}\big)$. However, since the KL-divergence between a Gaussian and a Gaussian mixture distributions does not have a closed-form expression, we estimate it using a slightly adapted method from (HO07). Our estimate is an average of the upper and lower bounds of the KL-divergence, denoted as $D_{\mathrm{var}}$ and $D_{\mathrm{prod}}$. Please refer to Appendix F for more details on this estimation. For better readability, we present the approximation of the KL-divergence by its upper bound $D_{\mathrm{var}}$ in the main part of this paper and we refer the reader to Appendix C for the approach using $\big(D_{\mathrm{var}} + D_{\mathrm{prod}}\big)/2$.

Finally, similar to (AFDM17; SZK23), we consider only the part of the upper bound MDL(**Q**) corresponding to the training dataset $S$, simply because the test dataset $S'$ is not available during the training phase. With this assumption and for a mini-batch $\mathcal{B} = \{z_1, \ldots, z_b\} \subseteq S$, the regularizer term is equal to

$$\mathrm{Regularizer}(\mathbf{Q}) := D_{KL}\big(P_{\mathbf{U}_\mathcal{B}|\mathbf{X}_\mathcal{B}, W_e} \big\| \mathbf{Q}_\mathcal{B}\big), \tag{17}$$

where the indices $\mathcal{B}$ indicate the restriction to the set $\mathcal{B}$. For better exposition, we will drop the notation dependence on $\mathcal{B}$ in the rest of this section. Now, we are ready to explain how the Gaussian mixtures are initialized, updated, and used as a regularizer simultaneously and along the optimization iterations. In what follows, the superscript $(t)$ denotes the optimization iteration $t \in \mathbb{N}^*$.

**Initialization.** First, we initialize the priors as $Q_c^{(0)}$ by initializing $\alpha_{c,m}^{(0)}$ and the parameters $\mu_{c,m}^{(0)}$ $\sigma_{c,m}^{(0)}$ of the components $Q_{c,m}^{(0)}$, for $c \in [C], m \in [M]$, similar to the method of initializing the centers in k-means++ (Art07). The reader is referred to Appendix C.1 for further details.

**Update of the priors.** Let the mini-batch picked at iteration $t$ be $\mathcal{B}^{(t)} = \{z_1^{(t)}, \ldots, z_b^{(t)}\}$. By dropping the dependence on $(t)$ for better readability, the regularizer 17, at iteration $(t)$, can be written as

$$\mathrm{Regularizer}(\mathbf{Q}) = \sum\nolimits_{i \in [b]} D_{KL}\big(P_{U_i|x_i,w_e} \big\| \sum\nolimits_{m \in [M]} \alpha_{y_i,m}^{(t)} Q_{y_i,m}^{(t)}(U_i)\big)$$

$$\stackrel{(a)}{\leqslant} \sum_{i \in [b]} \sum_{m \in [M]} \gamma_{i,m}\Big(D_{KL}\big(P_{U_i|x_i,w_e} \big\| Q_{y_i,m}^{(t)}(U_i)\big) - \log\big(\alpha_{y_i,m}^{(t)}/\gamma_{i,m}\big)\Big), \tag{18}$$

where the last step holds for any choices of $\gamma_{i,m} \geqslant 0$ such that $\sum_{m \in [M]} \gamma_{i,m} = 1$, for every $i \in [b]$. To see why the step $(a)$ holds, we refer the reader to Appendix F to see how the variational bound $D_{var}$ is derived.

Now, to update the components of the priors, first (similar to 'E'-step) note that the coefficients $\gamma_{i,m}$ that minimizes the above upper bound are equal to

$$\gamma_{i,m} = \frac{\alpha_{y_i,m}^{(t)} e^{-D_{KL}\big(P_{U_i|x_i,w_e} \| Q_{y_i,m}^{(t)}\big)}}{\sum_{m' \in [M]} \alpha_{y_i,m'}^{(t)} e^{-D_{KL}\big(P_{U_i|x_i,w_e} \| Q_{y_i,m'}^{(t)}\big)}}, \quad i \in [b], m \in [M]. \tag{19}$$

---

[2]While the bounds of Theorems 1 and 2 require the prior **Q** to satisfy the exact symmetry of Definition 1, it can be shown that these bounds still hold (with a small penalty) if such exact symmetry requirement is relaxed partially. The reader is referred to Appendix B, where formal results and their proofs are provided for the case of "almost symmetric" priors.

Let $\gamma_{i,c,m} = \gamma_{i,m}$ if $c = y_i$ and $\gamma_{i,c,m} = 0$ otherwise. Next, (similar to $M$-step) we treat $\gamma_{i,m}$ as constants, and find the parameters $\mu_{c,m}^*$, $\sigma_{c,m}^*$, $\alpha_{c,m}^*$ that minimizes the upper bound (18), by simply taking the partial derivatives and equating them to zero. Simple calculations show that the closed-form solutions are

$$\mu_{c,m}^* = \frac{1}{b_{c,m}} \sum_{i \in [b]} \gamma_{i,c,m} \mu_{x_i}, \quad \sigma_{c,m,j}^*{}^2 = \frac{1}{b_{c,m}} \sum_{i \in [b]} \gamma_{i,c,m} \left( \sigma_{x_i,j}^2 + (\mu_{x_i,j} - \mu_{c,m,j}^{(t)})^2 \right),$$

$$\alpha_{c,m}^* = b_{c,m}/b_c, \qquad\qquad b_{c,m} = \sum_{i \in [b]} \gamma_{i,c,m}, \qquad b_c = \sum_{m \in [M]} b_{c,m}. \qquad (20)$$

where $j \in [d]$ denotes the index of the coordinate in $\mathbb{R}^d$ and $\sigma_{c,m}^* = (\sigma_{c,m,1}^*, \dots, \sigma_{c,m,d}^*)$. Finally, to reduce the dependence of the prior on the dataset and to *partially* preserve the symmetry property, let

$$\mu_{c,m}^{(t+1)} = (1 - \eta_1)\mu_{c,m}^{(t)} + \eta_1 \mu_{c,m}^* + \mathfrak{Z}_1^{(t+1)}, \quad \sigma_{c,m}^{(t+1)}{}^2 = (1 - \eta_2)\sigma_{c,m}^{(t)}{}^2 + \eta_2 \sigma_{c,m}^*{}^2 + \mathfrak{Z}_2^{(t+1)},$$

$$\alpha_{c,m}^{(t+1)} = (1 - \eta_3)\alpha_{c,m}^{(t)} + \eta_3 \alpha_{c,m}^*, \qquad (21)$$

where $\eta_1, \eta_2, \eta_3 \in [0,1]$ are some fixed coefficients and $\mathfrak{Z}_j^{(t+1)}$, $j \in [2]$, are i.i.d. multivariate Gaussian random variables distributed as $\mathcal{N}(\mathbf{0}_d, \zeta_j^{(t+1)} I_d)$. Here $\mathbf{0}_d = (0, \dots, 0) \in \mathbb{R}^d$ and $\zeta_j^{(t+1)} \in \mathbb{R}^+$ are some fixed constants.

**Regularizer.** Finally, using (19), the upper bound (18) that we use as a regularizer can be simplified as

$$-\sum_{i \in [b]} \log \left( \sum_{m \in [M]} \alpha_{y_i,m}^{(t)} e^{-D_{KL}\left( P_{U_i|x_i,w_e} \| Q_{y_i,m}^{(t)} \right)} \right). \qquad (22)$$

## 4.2 Lossy Gaussian Mixture Prior

The lossy case is explained in Appendix C.3 when the KL-divergence estimate $(D_{\mathrm{prod}} + D_{\mathrm{var}})/2$ is considered. Similar to Section 4.1, it can be shown that if only $D_{\mathrm{var}}$ is considered for the KL-divergence estimate, then the regularizer term becomes equal to

$$-\sum_{i \in [b]} \log \left( \sum_{m \in [M]} \alpha_{y_i,m}^{(t)} e^{-D_{KL,Lossy}\left( P_{U_i|x_i,\hat{w}_e} \| Q_{y_i,m}^{(t)} \right)} \right), \qquad (23)$$

where $D_{KL,Lossy}\left( P_{U|x,\hat{w}_e} \| Q_{y,m} \right)$ is defined as

$$D_{KL}\left( \mathcal{N}\left( \mu_x, \frac{\sqrt{d}}{2} I_d \right) \Big\| \mathcal{N}\left( \mu_{c,m}, \frac{\sqrt{d}}{2} I_d \right) \right) + D_{KL}\left( \mathcal{N}\left( \mathbf{0}_d, \mathrm{diag}(\sigma_x^2 + \boldsymbol{\epsilon}) \right) \Big\| \mathcal{N}\left( \mathbf{0}_d, \mathrm{diag}(\sigma_{c,m}^2 + \boldsymbol{\epsilon}) \right) \right), \qquad (24)$$

where $\boldsymbol{\epsilon} = (\epsilon, \dots, \epsilon) \in \mathbb{R}^d$ and $\epsilon \in \mathbb{R}^+$ is a fixed hyperparameter.

Furthermore the components are updated according to (21), where $\gamma_{i,c,m}$, $\mu_{c,m}^*$, and $\alpha_{c,m}^*$ are defined as before, but $\sigma_{c,m,j}^*{}^2 = \frac{1}{b_{c,m}} \sum_{i \in [b]} \gamma_{i,c,m} \sigma_{x_i,j}^2$ and $\gamma_{i,m}$ is equal to

$$\gamma_{i,m} = \frac{\alpha_{y_i,m}^{(t)} e^{-D_{KL,Lossy}\left( P_{U_i|x_i,\hat{w}_e} \| Q_{y_i,m}^{(t)} \right)}}{\sum_{m' \in [M]} \alpha_{y_i,m'}^{(t)} e^{-D_{KL,Lossy}\left( P_{U_i|x_i,\hat{w}_e} \| Q_{y_i,m'}^{(t)} \right)}} = \frac{\beta_{y_i,m}^{(t)} e^{\frac{\langle \mu_{x_i}, \mu_{y_i,m}^{(t)} \rangle}{\sqrt{d}}}}{\sum_{m' \in [M]} \beta_{y_i,m'}^{(t)} e^{\frac{\langle \mu_{x_i}, \mu_{y_i,m'}^{(t)} \rangle}{\sqrt{d}}}},$$

where $\beta_{y_i,m}^{(t)} = \alpha_{y_i,m}^{(t)} e^{-\frac{\|\mu_{y_i,m}^{(t)}\|^2}{\sqrt{d}}} e^{-\sum_{j \in [d]}\left( \log(\sigma_{y_i,m,j}^{(t)}/\sigma_{x_i,j}) + \sigma_{x_i,j}^2/(2\sigma_{y_i,m,j}^{(t)}{}^2) \right)}$. In cases where the means of the components are normalized and the variances are fixed, $\beta_{y_i,m}^{(t)} \propto \alpha_{y_i,m}^{(t)}$.

The parameters $\gamma_{i,m}$ measure the contribution of the component $m$ in $Q_{y_i}$ in generating the latent variable $U_i$. One can observe a similarity between how these parameters are chosen in our approach and the attention mechanism, with the difference that here we are considering a *weighted* version of this mechanism, and without key and query matrices since we do not consider projections to other spaces. Intuitively, to measure the contribution of each component, we measure how much that component "*attend*" to $U_i$.

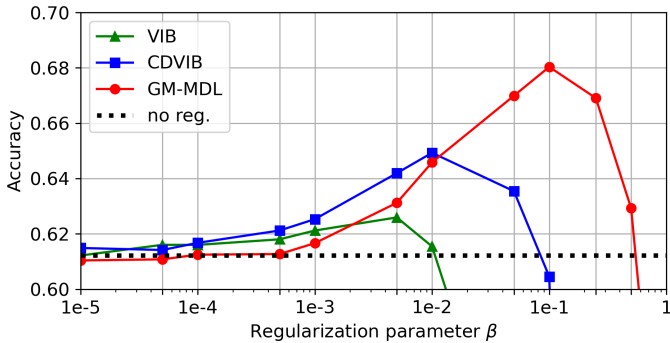

Figure 4: Test performance of the CNN-based encoder trained on CIFAR10 using standard VIB (AFDM17) regularization, Category-dependent VIB (CDVIB) (SZK23) regularization, and our proposed Gaussian Mixture MDL (GM-MDL) regularization.

## 5 EXPERIMENTS

In this section, we present the results of our simulations. The reader is referred to Appendix E for additional details, including used datasets, models, and training hyperparameters.

For the experiments, we considered the lossy regularizer approach with a Gaussian mixture prior and the KL-divergence estimate of $(D_{\text{prod}} + D_{\text{var}})/2$, as detailed in Appendix C.3. In this section, we refer to our regularizer as *Gaussian mixture MDL* (GM-MDL). To verify the practical benefits of the introduced regularizer, we conducted several experiments considering different datasets and encoder architectures as summarized below and detailed in Appendix E:

- **Datasets:** CIFAR10, CIFAR100, INTEL, and USPS image classification,

- **Encoder architectures:** CNN4 and ResNet18.

To compare our approach with the previous literature, in addition to the no-regularizer case, we also considered the Variational Information Bottleneck (VIB) of (AFDM17) and the Category-dependent VIB (CDVIB) of (SZK23).

The results presented in Fig. 4 and Table 1 clearly show the practical advantages of our proposed approach. All experiments are run independently 5 times and the reported values and plots are the average over 5 runs. In Fig.4, we plotted the performance of different regularizers as a function of the trade-off regularization parameter $\beta$. In Table 1, we reported the best achieved average test accuracy for each regularizer.

Table 1: Test performance of representation learning models with different encoder architectures, and trained on selected datasets using VIB (AFDM17), Category-dependent VIB (CDVIB) (SZK23), and our proposed Gaussian Mixture MDL (GM-MDL).

| # | Encoder | Dataset | no reg. | VIB | CDVIB | GM-MDL |
|---|---------|---------|---------|-----|-------|--------|
| 1 | CNN4 | CIFAR10 | 0.612 | 0.626 | 0.649 | **0.681** |
| 2 | CNN4 | USPS | 0.948 | 0.952 | 0.955 | **0.963** |
| 3 | CNN4 | INTEL | 0.756 | 0.759 | 0.763 | **0.776** |
| 4 | ResNet18 | CIFAR10 | 0.824 | 0.829 | 0.835 | **0.848** |
| 5 | ResNet18 | CIFAR100 | 0.454 | 0.458 | 0.463 | **0.497** |

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

# Appendices

The appendices are organized as follows:

## A    INTUITION BEHIND LOSSY GENERALIZATION BOUNDS

The bounds of Theorems 1 and 2 for the deterministic encoders may become vacuous due to the KL-divergence term, and the bounds cannot explain the empirically observed *geometrical compression* phenomenon (Gei21). These issues can be addressed using the *lossy* compressibility approach, as opposed to the *lossless* compressibility approach considered in previous sections. To provide a better intuition for these approaches, we first briefly explain their counterparts in information theory, i.e., lossless and lossy source compression.

Consider a *discrete* source $V \sim P_V$ and assume that we have $n$ i.i.d. realizations $V_1, \ldots, V_n$ of this source. Then, for sufficiently large values of $n$, the classical lossless source coding result in information theory states that this sequence can be described by approximately $nH(V)$ bits, where $H(V)$ is the Shannon entropy function. Thus, intuitively, $H(V)$ is the complexity of the source $V$. Now, suppose that $V$ is no longer discrete. Then $V_1, \ldots, V_n$ can no longer be described by any *finite* number of bits. However, if we consider some "vector quantization" instead, a sufficiently close vector can be described by a finite number of bits. This concept is called *lossy compression*. The amount of closeness is called the distortion, and the minimum number of needed bits (per sample) to describe the source within a given distortion level is given by the rate-distortion function.

Similar to (SZK23, Section 2.2.1 and Appendix C.1.2), we borrow such concepts to capture the "lossy complexity" of the latent variables in order to avoid non-vacuous bounds, which can also explain the geometrical compression phenomenon (Gei21; SZK23). This is achieved by considering the compressibility of "quantized" latent variables derived using the "distorted" encoders $\hat{W}_e$. Note that $\hat{W}_e$ is distorted only for the regularization term to measure the lossy compressibility (rate-distortion), and the undistorted latent variables are passed to the decoder. This is different from approaches that simply add noise to the output of the encoder and pass it to the decoder.

Finally, we show how to derive similar lossy bounds to (14) in terms of the function $h_D$. We first define the inverse of the function $h_D$ as follows. For any $y \in [0, 2]$ and $x_2 \in [0, 1]$, let

$$h_D^{-1}(y|x_2) = \sup\{x_1 \in [0, 1] \colon h_D(x_1, x_2) \leqslant y\}. \tag{25}$$

Let $\hat{W}_e \in \mathcal{W}_e$ be any quantized model defined by $P_{\hat{W}_e|S}$, that satisfy the *distortion* criterion $\mathbb{E}_{P_{S,W} P_{\hat{W}_e|S}} \Big[ \mathrm{gen}(S, W) - \mathrm{gen}(S, \hat{W}) \Big] \leqslant \epsilon$, where $\hat{W} = (\hat{W}_e, W_d)$. Then, using Theorem 1 for the

quantized model, we have

$$\mathbb{E}_{\mathbf{S},\mathbf{S}',\hat{W},\hat{\mathbf{Y}},\hat{\mathbf{Y}}'}\Big[h_D\Big(\hat{\mathcal{L}}(\mathbf{Y}',\hat{\mathbf{Y}}'),\hat{\mathcal{L}}(\mathbf{Y},\hat{\mathbf{Y}})\Big)\Big]\leqslant$$
$$\frac{\mathrm{MDL}(\mathbf{Q})+\log(n)}{n}+\mathbb{E}_{\mathbf{Y},\mathbf{Y}',\hat{\mathbf{Y}},\hat{\mathbf{Y}}'}\Big[h_{\mathbf{Y},\mathbf{Y}',\hat{\mathbf{Y}},\hat{\mathbf{Y}}'}\Big(\frac{1}{2}\|\hat{p}_{\mathbf{Y}}-\hat{p}_{\mathbf{Y}'}\|_1\Big)\Big]=:\Delta(\hat{W},\mathbf{Q}). \tag{26}$$

Next, using the Jensen inequality, we have

$$h_D\Big(\mathbb{E}_{\hat{W}}[\mathcal{L}(\hat{W})],\mathbb{E}_{S,\hat{W}}[\hat{\mathcal{L}}(S,\hat{W})]\Big)\Big]\leqslant\mathbb{E}_{\mathbf{S},\mathbf{S}',\hat{\mathbf{Y}},\hat{\mathbf{Y}}'}\Big[h_D\Big(\hat{\mathcal{L}}(\mathbf{Y}',\hat{\mathbf{Y}}'),\hat{\mathcal{L}}(\mathbf{Y},\hat{\mathbf{Y}})\Big)\Big]. \tag{27}$$

Combining the above two inequalities yields

$$h_D\Big(\mathbb{E}_{\hat{W}}[\mathcal{L}(\hat{W})],\mathbb{E}_{S,\hat{W}}[\hat{\mathcal{L}}(S,\hat{W})]\Big)\Big]\leqslant\Delta(\hat{W},\mathbf{Q}). \tag{28}$$

Finally, we have

$$\begin{aligned}\mathbb{E}_{\mathbf{S},W}[\mathrm{gen}(S,W)]&\leqslant\mathbb{E}_{\mathbf{S},\hat{W}}\Big[\mathrm{gen}(S,\hat{W})\Big]+\epsilon\\&=\mathbb{E}_{\hat{W}}[\mathcal{L}(\hat{W})]-\mathbb{E}_{S,\hat{W}}[\hat{\mathcal{L}}(S,\hat{W})]+\epsilon\\&\leqslant h_D^{-1}\Big(\min(2,\Delta(\hat{W},\mathbf{Q}))|\mathbb{E}_{S,\hat{W}}[\hat{\mathcal{L}}(S,\hat{W})]\Big)-\mathbb{E}_{S,\hat{W}}[\hat{\mathcal{L}}(S,\hat{W})]+\epsilon\end{aligned} \tag{29}$$

In particular, for negligible values of $\mathbb{E}_{S,\hat{W}}[\hat{\mathcal{L}}(S,\hat{W})]$, $h_D^{-1}\Big(\min(2,\Delta(\hat{W},\mathbf{Q}))|\mathbb{E}_{S,\hat{W}}[\hat{\mathcal{L}}(S,\hat{W})]\Big)\approx$ $\min(2,\Delta(\hat{W},\mathbf{Q}))\lesssim\frac{\mathrm{MDL}(\mathbf{Q})+\log(n)}{n}$, which gives

$$\mathbb{E}_{\mathbf{S},W}[\mathrm{gen}(S,W)]\lesssim\frac{\mathrm{MDL}(\mathbf{Q})+\log(n)}{n}+\epsilon.$$

# B GENERALIZATION BOUNDS VIA NON-SYMMETRIC PRIORS

In this section, we discuss how the bounds of Theorems 1 and 2 can be extended to settings in which the requirement of symmetry is relaxed partially. We focus on "differentially private" and "partially symmetric" data-dependent priors.

## B.1 DIFFERENTIALLY PRIVATE DATA-DEPENDENT PRIORS

One way to extend the results to include the partially symmetric data-dependent priors is by leveraging the differential privacy tools (Dwo06; DR+14; DFH+15; DR18). The reader is referred to (Alq21, Section 3.3) for more on differentially private priors.

Recall that given the dataset $S$ we train a model $W$ using the learning algorithm $\mathcal{A}(\cdot)$, *i.e.*, $W=\mathcal{A}(S)$. Now, assume that by having the dataset $S$ and the trained model $W=\mathcal{A}(S)$ we choose the prior $\mathbf{Q}^{S,W}$ using a potentially stochastic mechanism $\mathcal{T}\colon\mathcal{S}\times\mathcal{W}\to\mathcal{Q}$, where $\mathcal{Q}$ denotes the space of all conditional distributions of $\mathbf{U},\mathbf{U}'$ given $(\mathbf{Y},\mathbf{Y}')$, that is "strongly" symmetric. To state the definition of strongly symmetric prior, we first recall the notations of $\mathbf{U}_\pi,\mathbf{U}'_\pi$ and $\mathbf{Y}_\pi,\mathbf{Y}'_\pi$ for any permutation $\pi\colon[2n]\to[2n]$. Let $Y^{2n}:=(\mathbf{Y},\mathbf{Y}')$. Then, we define $\mathbf{Y}_\pi$ and $\mathbf{Y}'_\pi$ as

$$\begin{aligned}\mathbf{Y}_\pi&:=Y_{\pi(1)},\ldots,Y_{\pi(n)},\\\mathbf{Y}_\pi&:=Y_{\pi(n+1)},\ldots,Y_{\pi(2n)}.\end{aligned} \tag{30}$$

The variables $\mathbf{U}_\pi$ and $\mathbf{U}'_\pi$ are defined in a similar manner.

**Definition 2** (Strongly symmetric prior). *A conditional distribution $\mathbf{Q}$ of $\mathbf{U},\mathbf{U}'$ given $(\mathbf{Y},\mathbf{Y}')$ is strongly symmetric, if for every $(\mathbf{U},\mathbf{U}',\mathbf{Y},\mathbf{Y}')$ and every permutation $\pi\colon[2n]\to[2n]$ that preserves the labeling (i.e., $\mathbf{Y}_\pi=\mathbf{Y}$ and $\mathbf{Y}'_\pi=\mathbf{Y}'$) we have*

$$\mathbf{Q}\big(\mathbf{U},\mathbf{U}'|\mathbf{Y},\mathbf{Y}'\big)=\mathbf{Q}\big(\mathbf{U}_\pi,\mathbf{U}'_\pi|\mathbf{Y},\mathbf{Y}'\big). \tag{31}$$

Note that any strongly symmetric prior satisfies the symmetry condition of Definition 1. In addition, the per-label Gaussian-mixture prior of Section 4 meets the strongly symmetric condition. To show this, recall that for any $c \in [C]$, the Gaussian mixture prior for label $c$ is denoted by $Q_c$. Given these per-label priors, the prior $\mathbf{Q}$ is defined as

$$
\begin{aligned}
\mathbf{Q}(\mathbf{U}, \mathbf{U}'|S, S', \hat{W}_e) &= \mathbf{Q}(\mathbf{U}, \mathbf{U}'|\mathbf{Y}, \mathbf{Y}') \\
&= \prod_{i \in [n]} Q_{Y_i}(U_i) Q_{Y_i'}(U_i').
\end{aligned}
$$

It is immediate to see that this prior is strongly symmetric under any permutation that preserves the labeling.

Next, we define the notion of learning the prior in a differentially private manner. For simplicity, we consider the case where $\mathcal{A}(S)$ can be written as a deterministic function $g(S, V)$, where $V$ represents all the stochasticity in the learning algorithm that is independent of $S$. An example of such a learning algorithm is the Stochastic Gradient Descent (SGD) algorithm.

**Definition 3** (Differentially private prior). *We say $\mathcal{T} \colon \mathcal{S} \times \mathcal{W} \to \mathcal{Q}$ is $\varepsilon_p$-differentially private if for any fixed $V$, and all datasets $S$ and $S_1$ that are different in only one coordinate and for all measurable subsets $B \subseteq \mathcal{Q}$, we have*

$$
\mathbb{P}\Big(\mathbf{Q}^{S, \mathcal{A}(S)} \in B\Big) \leqslant e^{\varepsilon_p} \mathbb{P}\Big(\mathbf{Q}^{S_1, \mathcal{A}(S_1)} \in B\Big), \tag{32}
$$

*where $\mathcal{A}(S) = g(S, V)$ and $\mathcal{A}(S_1) = g(S_1, V)$.*

Now, we state our tail-bound result for $\varepsilon_p$-differentially private prior.

**Proposition 1.** *Consider the setup of Theorem 1 and suppose the prior $\mathbf{Q}^{S, \mathcal{A}(S)}$ is chosen using an $\varepsilon_p$-differentially private mechanism $\mathcal{T} \colon \mathcal{S} \times \mathcal{W} \to \mathcal{Q}$. Then, for any $\delta \geqslant 0$ and for $n \geqslant 10$, with probability at least $1 - \delta$ over choices of $(S, S', W)$, it holds that*

$$
\begin{aligned}
h_D\Big(\hat{\mathcal{L}}(S', W), \hat{\mathcal{L}}(S, W)\Big) \leqslant{} & \frac{D_{KL}\big(P_{\mathbf{U}, \mathbf{U}'|\mathbf{X}, \mathbf{X}', W_e} \big\| \mathbf{Q}^{S, \mathcal{A}(S)}\big) + \log(2n/\delta)}{n} \\
& + \mathbb{E}_{\hat{\mathbf{Y}}, \hat{\mathbf{Y}}'|\mathbf{Y}, \mathbf{Y}'}\left[h_{\mathbf{Y}, \mathbf{Y}', \hat{\mathbf{Y}}, \hat{\mathbf{Y}}'}\left(\frac{1}{2}\|\hat{p}_{\mathbf{Y}} - \hat{p}_{\mathbf{Y}'}\|_1\right)\right] \\
& + \frac{1}{2}\varepsilon_p^2 + \varepsilon_p\sqrt{\frac{\log(4/\delta)}{2n}}, 
\end{aligned} \tag{33}
$$

*where $\hat{p}_{\mathbf{Y}}$ and $\hat{p}_{\mathbf{Y}'}$ are empirical distributions of $\mathbf{Y}$ and $\mathbf{Y}'$, respectively.*

The proof stated in Appendix G.3 is an extension of Theorem 2 using (DFH$^+$15, Theorems 18&19) and (DR18, Theorem 4.2).

## B.2 PARTIALLY SYMMETRIC DATA-DEPENDENT PRIORS

In this section, we show an alternative way to extend our generalization bound results by defining the partially symmetric priors.

**Definition 4** (Partially symmetric prior). *The prior $\mathbf{Q}$ is $(\epsilon, \delta)$-partially symmetric for the learning algorithm $\mathcal{A} \colon \mathcal{Z}^n \to \mathcal{W}$, if with probability at least $1 - \delta$ over choices of $(S', S, W_e, \mathbf{U}, \mathbf{U}') \sim P_{S'} P_{S, W_e} \mathbf{Q}$,*

$$
\forall \pi_{\mathbf{Y}, \mathbf{Y}'} \colon \quad \mathbf{Q}(\mathbf{U}, \mathbf{U}'|\mathbf{Y}, \mathbf{Y}', \mathbf{X}, \mathbf{X}', W_e) \leqslant e^{\epsilon} \mathbf{Q}(\mathbf{U}_{\pi_{\mathbf{Y}, \mathbf{Y}'}}, \mathbf{U}'_{\pi_{\mathbf{Y}, \mathbf{Y}'}}|\mathbf{Y}, \mathbf{Y}', \mathbf{X}, \mathbf{X}', W_e), \tag{34}
$$

*where this should hold for any permutation $\pi_{\mathbf{Y}, \mathbf{Y}'}$ (which could potentially depend on $\mathbf{Y}, \mathbf{Y}'$) that satisfies the labeling.*

Note that the partially symmetric prior can potentially depend on $(S, W)$.

**Proposition 2.** *Consider the setup of Theorem 1. Then, for any $(\epsilon, \delta)$-partially symmetric conditional distribution $\mathbf{Q}$ and for $n \geqslant 10$, we have*

$$
\begin{aligned}
\mathbb{E}_{\mathbf{S}, \mathbf{S}', W, \hat{\mathbf{Y}}, \hat{\mathbf{Y}}'}\Big[h_D\Big(\hat{\mathcal{L}}(\mathbf{Y}', \hat{\mathbf{Y}}'), \hat{\mathcal{L}}(\mathbf{Y}, \hat{\mathbf{Y}})\Big)\Big] \leqslant{} & \frac{\mathrm{MDL}(\mathbf{Q}) + \log\big(\delta e^{2n} + n e^{\epsilon}\big)}{n} \\
& + \mathbb{E}_{\mathbf{Y}, \mathbf{Y}', \hat{\mathbf{Y}}, \hat{\mathbf{Y}}'}\left[h_{\mathbf{Y}, \mathbf{Y}', \hat{\mathbf{Y}}, \hat{\mathbf{Y}}'}\left(\frac{1}{2}\|\hat{p}_{\mathbf{Y}} - \hat{p}_{\mathbf{Y}'}\|_1\right)\right].
\end{aligned} \tag{35}
$$

*where $\hat{p}_{\mathbf{Y}}$ and $\hat{p}_{\mathbf{Y}'}$ are empirical distributions of $\mathbf{Y}$ and $\mathbf{Y}'$, respectively and* $\mathrm{MDL}(\mathbf{Q})$ *is defined in 10.*

This result is proved in Appendix G.4.

## C  GAUSSIAN MIXTURE PRIOR APPROXIMATION AND REGULARIZATION

In this section, we explain in detail our approach to finding an appropriate data-dependent Gaussian mixture prior and how to use it in a regularizer term along the optimization trajectories. The section is subdivided into three parts: the first part explains how we initialize the components of the Gaussian mixture prior, and the other two parts explain the lossless and lossy versions of our approach.

Recall that we are considering a regularizer term equal to

$$\mathrm{Regularizer}(\mathbf{Q}) := D_{KL}\big(P_{\mathbf{U}_{\mathcal{B}}|\mathbf{X}_{\beta},W_e}\big\|\mathbf{Q}_{\mathcal{B}}\big), \tag{36}$$

where the indices $\mathcal{B}$ indicate the restriction to the set $\mathcal{B}$. However, for the sake of simplicity, we will drop the dependence on $\mathcal{B}$ in the rest of this section. Also, in the following, the superscript $(t)$ is used to denote the optimization iteration $t \in \mathbb{N}^*$.

We choose a Gaussian mixture prior $\mathbf{Q}$ in lossless and lossy ways. In both approaches, we initialize three sets of parameters $\alpha_{c,m}^{(0)}$, $\mu_{c,m}^{(0)}$, and $\sigma_{c,m}^{(0)}$, for $c \in [C]$ and $m \in [M]$, similarly. We will explain this first.

### C.1  INITIALIZATION OF THE COMPONENTS

We let $\alpha_{c,m}^{(0)} = 1/M$, for $c \in [C]$ and $m \in [M]$. The standard deviation values $\sigma_{c,m}^{(0)}$ are randomly chosen from the distribution $\mathcal{N}(0, \mathrm{I}_d)$.

The means of the components $\mu_{c,m}^{(0)}$ are initialized in a way that the centers are initialized in the k-means++ method (Art07). More specifically, they are initialized as follows.

1. The model's encoder $W_e$ is initialized.
2. A mini-batch $\mathbf{Z} = \{Z_1, \ldots, Z_{\tilde{b}}\}$, with a large mini-batch size $\tilde{b} \gg b$, of the training data is selected. Let $\mathbf{X}$ and $\mathbf{Y}$ be the set of features and labels of this mini-batch.
For simplicity, we denote by $\mathbf{X}_c = \{X_{c,1}, \ldots, X_{c,b_c}\} \subseteq \mathbf{X}$ the subset of features of the mini-batch with label $c \in [C]$. Note that $\sum_{c \in [C]} b_c = \tilde{b}$.
Using the initialized encoder, we compute the corresponding parameters of the latent spaces for this mini-batch. Denote their mean vector as $\boldsymbol{\mu}_c = \{\mu_{c,1}, \ldots, \mu_{c,b_c}\}$. For each $c \in [C]$, we let $\mu_{c,1}^{(0)}$ be equal to one of the elements in $\boldsymbol{\mu}_c$, uniformly.
3. For $2 \leqslant m \leqslant M$, we take a new mini-batch $\mathbf{Z}$, with per-label features and latent variable means $\mathbf{X}_c$ and $\boldsymbol{\mu}_c$. Then, for all $c \in [C]$, we compute the below distances:

$$d_{\min,c}(i) = \min_{m' \in [m-1]} \left\| \mu_{c,i} - \mu_{c,m'}^{(0)} \right\|^2, \qquad i \in [b_c].$$

Then, we randomly sample an index $i^*$ from the set $[b_c]$ according to a weighted probability distribution, where the index $i$ has a weight proportional to $d_{\min,c}(i)$. We let $\mu_{c,m}^{(0)}$ be equal to $\mu_{c,i^*}$.

### C.2  LOSSLESS GAUSSIAN MIXTURE PRIOR

We start with the lossless version, which is easier to explain. Based on our observations in the experiments, the final population accuracy achieved when using the lossless regularizer is better than when using VIB (AFDM17) or CDVIB (SZK23) but worse than when using the lossy version, explained in Appendix C.3.

**Update of the priors.** Suppose the mini-batch picked at iteration $t$ is $\mathcal{B}^{(t)} = \{z_1^{(t)}, \ldots, z_b^{(t)}\}$. We drop the dependence of the samples on $(t)$ for better readability. Then, the regularizer (36), at iteration

$(t)$, can be written as

$$\text{Regularizer}(\mathbf{Q}) = \sum_{i \in [b]} D_{KL}\big(P_{U_i|x_i,w_e} \| Q_{y_i}^{(t)}(U_i)\big). \tag{37}$$

We propose upper and lower bounds on this term. The upper bound is already presented in (18), denoted as $D_{\text{var}}$:

$$\text{Regularizer}(\mathbf{Q}) \leqslant D_{\text{var}} := \sum_{i \in [b]} \sum_{m \in [M]} \gamma_{i,m} \left( D_{KL}\big(P_{U_i|x_i,w_e} \| Q_{y_i,m}^{(t)}(U_i)\big) - \log\left(\frac{\alpha_{y_i,m}^{(t)}}{\gamma_{i,m}}\right) \right). \tag{38}$$

The upper bound holds for all choices of $\gamma_{i,m} \geqslant 0$ such that $\sum_{m \in [M]} \gamma_{i,m} = 1$, for any $i \in [b]$. As explained in Section 4, the coefficients $\gamma_{i,m}$ that minimize the above upper bound and thus make it tighter are equal to

$$\gamma_{i,m} = \frac{\alpha_{y_i,m}^{(t)} e^{-D_{KL}\left(P_{U_i|x_i,w_e} \| Q_{y_i,m}^{(t)}\right)}}{\sum_{m' \in [M]} \alpha_{y_i,m'}^{(t)} e^{-D_{KL}\left(P_{U_i|x_i,w_e} \| Q_{y_i,m'}^{(t)}\right)}}, \quad i \in [b], m \in [M]. \tag{39}$$

Denote $\gamma_{i,c,m} = \begin{cases} \gamma_{i,m}, & \text{if } c = y_i, \\ 0, & \text{otherwise..} \end{cases}$

Next, we establish an estimated lower bound on the regularizer as

$$\text{Regularizer}(\mathbf{Q}) \geqslant -\sum_{i \in [b]} \left( \frac{1}{2} \log\left( (2\pi e)^d \prod_{j \in [d]} \sigma_{x_i,j}^2 \right) + \log\left( \sum_{m=1}^{M} \alpha_{y_i,m}^{(t)} t_{i,m} \right) \right)$$

$$\approx -\sum_{i \in [b]} \left( \frac{1}{2} \log\left( (2\pi e)^d \prod_{j \in [d]} \sigma_{x_i,j}^2 \right) + \log\left( \sum_{m=1}^{M} \alpha_{y_i,m}^{(t)} t'_{i,m} \right) \right)$$

$$=: D_{\text{prod}}, \tag{40}$$

where

$$t_{i,m} := \mathbb{E}_{U \sim P_{U_i|x_i,w_e}}\left[ Q_{y_i,m}^{(t)} \right] \overset{(a)}{=} \frac{e^{-\sum_{j \in [d]} \frac{\left(\mu_{x_i,j} - \mu_{y_i,m,j}^{(t)}\right)^2}{2\left(\sigma_{x_i,j}^2 + \sigma_{y_i,m,j}^{(t)}{}^2\right)}}}{\sqrt{\prod_{j \in [d]} \left( 2\pi \left( \sigma_{x_i,j}^2 + \sigma_{y_i,m,j}^{(t)}{}^2 \right) \right)}},$$

$$t'_{i,m} := \frac{e^{-\sum_{j \in [d]} \frac{\left(\mu_{x_i,j} - \mu_{y_i,m,j}^{(t)}\right)^2}{2\sigma_{y_i,m,j}^{(t)}{}^2}}}{\sqrt{\prod_{j \in [d]} \left( 2\pi \sigma_{y_i,m,j}^{(t)}{}^2 \right)}}, \tag{41}$$

where the step $(a)$ is derived from (Bro03). The reader is referred to Appendix F for details on how this upper bound is derived.

It has already been observed in (HO07) for the case of two Gaussian mixture distributions that the KL-divergence is better estimated by considering the average of the *product* lower bound and the *variational* upper bound. We then consider the following estimate as the regularizer term

$$\text{Regularizer}(\mathbf{Q}) \approx \frac{D_{\text{var}} + D_{\text{prod}}}{2} =: D_{\text{est}}, , \tag{42}$$

where $D_{\text{var}}$ and $D_{\text{prod}}$ are defined in (38) and (40), respectively.

Next, we treat $\gamma_{i,m}$ as constants and find the parameters $\mu_{c,m}^*$, $\sigma_{c,m}^*$, $\alpha_{c,m}^*$ that minimize $D_{\text{est}}$ by solving the following equations

$$\frac{\partial D_{est}}{\partial \mu_{c,m,j}} = 0, \quad \frac{\partial D_{est}}{\partial \sigma_{c,m,j}} = 0, \quad \frac{\partial D_{est}}{\partial \alpha_{c,m}} = 0,$$

with the constraint that $\sum_m \alpha_{c,m} = 1$ for each $c \in [C]$. The above equations have the following optimal solutions $\mu^*_{c,m}$ and $\alpha^*_{c,m}$, and $\sigma^*_{c,m}$:

$$\mu^*_{c,m} = \frac{1}{\tilde{b}_{c,m}} \sum\nolimits_{i \in [b]} \tilde{\gamma}_{i,c,m} \mu_{x_i},$$

$$\sigma^{*\,2}_{c,m,j} = \frac{1}{b_{c,m}} \sum\nolimits_{i \in [b]} \left( \gamma_{i,c,m} \sigma^2_{x_i,j} + 2\tilde{\gamma}_{i,c,m} (\mu_{x_i,j} - \mu^{(t)}_{c,m,j})^2 \right),$$

$$\alpha^*_{c,m} = \tilde{b}_{c,m}/\tilde{b}_c,$$

$$\tilde{b}_{c,m} = \sum\nolimits_{i \in [b]} \tilde{\gamma}_{i,c,m},$$

$$\tilde{b}_c = \sum\nolimits_{m \in [M]} \tilde{b}_{c,m}$$

$$b_{c,m} = \sum\nolimits_{i \in [b]} \gamma_{i,c,m}, . \tag{43}$$

where

$$\tilde{\gamma}_{i,c,m} := \frac{\gamma_{i,c,m} + \beta_{i,c,m}}{2},$$

$$\beta_{i,c,m} = \begin{cases} \frac{\eta_{i,m}}{\sum_{m' \in [M]} \eta_{i,m'}}, & if\ c = y_i, \\ 0, & otherwise. \end{cases}$$

$$\eta_{i,m} := \alpha^{(t)}_{y_i,m} e^{-\sum_{j \in [d]} \frac{\left(\mu_{x_i,j} - \mu^{(t)}_{y_i,m,j}\right)^2}{2\sigma^{(t)\,2}_{y_i,m,j}}}. \tag{44}$$

Note that $j \in [d]$ denotes the index of the coordinate in $\mathbb{R}^d$ and $\sigma^*_{c,m} = (\sigma^*_{c,m,1}, \ldots, \sigma^*_{c,m,d})$. Finally, to reduce the dependence of the prior on the dataset, we choose the updates as

$$\mu^{(t+1)}_{c,m} = (1 - \eta_1)\mu^{(t)}_{c,m} + \eta_1 \mu^*_{c,m} + \mathfrak{Z}^{(t+1)}_1, \quad \sigma^{(t+1)\,2}_{c,m} = (1 - \eta_2)\sigma^{(t)\,2}_{c,m} + \eta_2 \sigma^{*\,2}_{c,m} + \mathfrak{Z}^{(t+1)}_2,$$

$$\alpha^{(t+1)}_{c,m} = (1 - \eta_3)\alpha^{(t)}_{c,m} + \eta_3 \alpha^*_{c,m}, \tag{45}$$

where $\eta_1, \eta_2, \eta_3 \in [0,1]$ are some fixed coefficients and $\mathfrak{Z}^{(t+1)}_j$, $j \in [2]$, are i.i.d. multivariate Gaussian random variables distributed as $\mathcal{N}(\mathbf{0}_d, \zeta^{(t+1)}_j \mathrm{I}_d)$. Here $\mathbf{0}_d = (0, \ldots, 0) \in \mathbb{R}^d$ and $\zeta^{(t+1)}_j \in \mathbb{R}^+$ are some fixed constants.

**Regularizer.** Finally, the regularizer estimation (42) can be simplified as

$$\mathrm{Regularizer}(\mathbf{Q}) = -\frac{1}{2} \sum_{i \in [b]} \log \left( \sum\nolimits_{m \in [M]} \alpha^{(t)}_{y_i,m} e^{-D_{KL}\left(P_{U_i|x_i,w_e} \| Q^{(t)}_{y_i,m}\right)} \right)$$

$$-\frac{1}{2} \sum_{i \in [b]} \left( \frac{1}{2} \log \left( (2\pi e)^d \prod\nolimits_{j \in [d]} \sigma^2_{x_i,j} \right) + \log \left( \sum_{m=1}^{M} \alpha^{(t)}_{y_i,m} t'_{i,m} \right) \right). \tag{46}$$

## C.3 Lossy Gaussian mixture prior

Now, we proceed with the lossy version of the regularizer. For this, we consider the MDL of the "perturbed" latent variable while passing the unperturbed latent variable to the decoder. Fix some $\epsilon \in \mathbb{R}^+$ and let $\boldsymbol{\epsilon} = (\epsilon, \ldots, \epsilon) \in \mathbb{R}^d$.

For the regularizer, we first consider the perturbed $U$ as

$$\hat{U} = U + \tilde{Z} = (\mu_X + Z_1) + Z_2 =: \hat{U}_1 + \hat{U}_2, \tag{47}$$

where $\tilde{Z}$, $Z_1$, and $Z_2$ are independent multi-variate random variables, drawn from the distributions $\mathcal{N}\left(\mathbf{0}_d, \sqrt{d/4}\,\mathrm{I}_d + \mathrm{diag}\left(\boldsymbol{\epsilon}\right)\right)$, $\mathcal{N}\left(\mathbf{0}_d, \sqrt{d/4}\,\mathrm{I}_d\right)$, and $\mathcal{N}\left(\mathbf{0}_d, \mathrm{diag}\left(\sigma^2_{X,j} + \epsilon\right)\right)$, respectively. Consequently, $\hat{U}_1 \sim \mathcal{N}(\mu_X, \sqrt{d/4}\,\mathrm{I}_d)$ is independent from $\hat{U}_2 \sim \mathcal{N}(\mathbf{0}_d, \mathrm{diag}(\sigma^2_X + \boldsymbol{\epsilon}))$, given $(X, W_e)$.

For each label $c \in [C]$, we consider two Gaussian mixture priors $Q_{c,1}$ and $Q_{c,2}$ for $\hat{U}_1$ and $\hat{U}_2$, respectively, as follows:

$$Q_{c,1} = \sum_{m \in [M]} \alpha_{c,m} Q_{c,m,1}, \tag{48}$$

$$Q_{c,2} = \sum_{m \in [M]} \alpha_{c,m} Q_{c,m,2} \tag{49}$$

over $\mathbb{R}^d$, where $\alpha_{c,m} \in [0,1]$, $\sum_{m \in [M]} \alpha_{c,m} = 1$ for each $c \in [C]$, and where $\{Q_{c,m,1}\}_{c,m}$ and $\{Q_{c,m,2}\}_{c,m}$ are multivariate Gaussian distributions with a diagonal covariance matrix:

$$Q_{c,m,1} = \mathcal{N}\left(\mu_{c,m}, \sqrt{d/4}\,\mathrm{I}_d\right),$$

$$Q_{c,m,2} = \mathcal{N}\left(\mathbf{0}_d, \mathrm{diag}\left(\sigma_{c,m}^2 + \boldsymbol{\epsilon}\right)\right).$$

Note that the Gaussian mixture priors $Q_{c,1}$ and $Q_{c,2}$ have the same parameters of $\alpha_{c,m}$. Now, let the prior $Q_c$ be the distortion of $\hat{U} = \hat{U}_1 + \hat{U}_2$, when $\hat{U}_1 \sim Q_{c,1}$ and $\hat{U}_2 \sim Q_{c,2}$.

Now, for the variation upper bound $D_{\mathrm{var}}$ for the regularizer, we first consider the inequality

$$D_{KL}\left(P_{\hat{U}|x,w_e}\|Q_{y_i}\right) \leq D_{KL}\left(\mathcal{N}(\mu_x, \sqrt{d/4}\,\mathrm{I}_d)\|Q_{y_i,1}\right) + D_{KL}\left(\mathcal{N}(\mathbf{0}_d, \mathrm{diag}(\sigma_x^2 + \boldsymbol{\epsilon}))\|Q_{y_i,2}\right)$$

$$=: D_{KL,Lossy}\left(P_{\hat{U}|x,w_e}\|Q_{y_i}\right). \tag{50}$$

Using the same arguments as in the lossless version but for $D_{KL,Lossy}\left(P_{\hat{U}|x,w_e}\|Q_{y_i}\right)$ instead of $D_{KL}\left(P_{\hat{U}|x,w_e}\|Q_{y_i}\right)$, we derive the following upper bound, denoted as $D_{\mathrm{var}}$:

$$\mathrm{Regularizer}(\mathbf{Q}) \leq D_{\mathrm{var}} := \sum_{i \in [b]} \sum_{m \in [M]} \gamma_{i,m} \left( D_{KL,lossy}\left(P_{U_i|x_i,w_e}\|Q_{y_i,m}^{(t)}(U_i)\right) - \log\left(\frac{\alpha_{y_i,m}^{(t)}}{\gamma_{i,m}}\right) \right), \tag{51}$$

which is minimized for

$$\gamma_{i,m} = \frac{\alpha_{y_i,m}^{(t)} e^{-D_{KL,Lossy}\left(P_{U_i|x_i,w_e}\|Q_{y_i,m}^{(t)}\right)}}{\sum_{m' \in [M]} \alpha_{y_i,m'}^{(t)} e^{-D_{KL,Lossy}\left(P_{U_i|x_i,w_e}\|Q_{y_i,m'}^{(t)}\right)}}, \quad i \in [b], m \in [M]. \tag{52}$$

Denote $\gamma_{i,c,m} = \begin{cases} \gamma_{i,m}, & \text{if } c = y_i, \\ 0, & \text{otherwise.} \end{cases}$

For the lower bound, we apply a similar lower bound as in the lossless case. This (estimated) lower bound, denoted by $D_{\mathrm{prod}}$, is equal to

$$D_{\mathrm{prod}} := -\sum_{i \in [b]} \left( \frac{d}{2} \log\left(\pi e \sqrt{d}\right) + \log\left( \sum_{m=1}^{M} \alpha_{y_i,m}^{(t)} \tilde{t}_{i,m} \right) \right), \tag{53}$$

where

$$\tilde{t}_{i,m} := \frac{1}{\sqrt{(2\pi\sqrt{d})^d}} e^{-\frac{\left\| \mu_{x_i} - \mu_{y_i,m}^{(t)} \right\|^2}{2\sqrt{d}}}, \tag{54}$$

We then consider the following estimate as the regularizer term

$$\mathrm{Regularizer}(\mathbf{Q}) \approx \frac{D_{\mathrm{var}} + D_{\mathrm{prod}}}{2} =: D_{\mathrm{est}}, \tag{55}$$

where $D_{\mathrm{var}}$ and $D_{\mathrm{prod}}$ are defined in (51) and (53), respectively.

Next, similar to the lossless case, we treat $\gamma_{i,m}$ as constants and find the parameters $\mu_{c,m}^*, \sigma_{c,m}^*, \alpha_{c,m}^*$ that minimize $D_{\mathrm{est}}$ by solving the following equations

$$\frac{\partial D_{est}}{\partial \mu_{c,m,j}} = 0, \quad \frac{\partial D_{est}}{\partial \sigma_{c,m,j}} = 0, \quad \frac{\partial D_{est}}{\partial \alpha_{c,m}} = 0,$$

with the constraint that $\sum_m \alpha_{c,m} = 1$ for each $c \in [C]$. The exact closed-form solutions $\mu_{c,m}^*$ and $\alpha_{c,m}^*$ and $\sigma_{c,m,j}^*$ are equal to :

$$\mu_{c,m}^* = \frac{1}{\hat{b}_{c,m}} \sum\nolimits_{i \in [b]} \hat{\gamma}_{i,c,m} \mu_{x_i},$$

$$\sigma_{c,m,j}^{*2} = \frac{1}{b_{c,m}} \sum\nolimits_{i \in [b]} \gamma_{i,c,m} \sigma_{x_i,j}^2,$$

$$\alpha_{c,m}^* = \tilde{b}_{c,m} / \tilde{b}_c,$$

$$\tilde{b}_{c,m} = \sum\nolimits_{i \in [b]} \tilde{\gamma}_{i,c,m},$$

$$\tilde{b}_c = \sum\nolimits_{m \in [M]} \tilde{b}_{c,m}$$

$$b_{c,m} = \sum\nolimits_{i \in [b]} \gamma_{i,c,m},$$

$$\hat{b}_{c,m} = \sum\nolimits_{i \in [b]} \hat{\gamma}_{i,c,m}. \tag{56}$$

where

$$\tilde{\gamma}_{i,c,m} := \frac{\gamma_{i,c,m} + \beta_{i,c,m}}{2},$$

$$\hat{\gamma}_{i,c,m} := \frac{2\gamma_{i,c,m} + \beta_{i,c,m}}{3},$$

$$\beta_{i,c,m} = \begin{cases} \frac{\eta_{i,m}}{\sum_{m' \in [M]} \eta_{i,m'}}, & if\ c = y_i, \\ 0, & otherwise. \end{cases}$$

$$\eta_{i,m} := \alpha_{y_i,m}^{(t)} e^{-\frac{\left\| \mu_{x_i} - \mu_{y_i,m}^{(t)} \right\|^2}{2\sqrt{d}}}. \tag{57}$$

Note that $j \in [d]$ denotes the index of the coordinate in $\mathbb{R}^d$ and $\sigma_{c,m}^* = (\sigma_{c,m,1}^*, \ldots, \sigma_{c,m,d}^*)$. Finally, to reduce the dependence of the prior on the dataset, we choose the updates

$$\mu_{c,m}^{(t+1)} = (1 - \eta_1)\mu_{c,m}^{(t)} + \eta_1 \mu_{c,m}^* + \mathfrak{Z}_1^{(t+1)}, \quad \sigma_{c,m}^{(t+1)2} = (1 - \eta_2)\sigma_{c,m}^{(t)2} + \eta_2 \sigma_{c,m}^{*2} + \mathfrak{Z}_2^{(t+1)},$$

$$\alpha_{c,m}^{(t+1)} = (1 - \eta_3)\alpha_{c,m}^{(t)} + \eta_3 \alpha_{c,m}^*, \tag{58}$$

where $\eta_1, \eta_2, \eta_3 \in [0,1]$ are some fixed coefficients and $\mathfrak{Z}_j^{(t+1)}$, $j \in [2]$, are i.i.d. multivariate Gaussian random variables distributed as $\mathcal{N}(\mathbf{0}_d, \zeta_j^{(t+1)}\mathbf{I}_d)$. Here $\zeta_j^{(t+1)} \in \mathbb{R}^+$ are some fixed constants.

**Regularizer.** Finally, the regularizer estimation (55) can be simplified as

$$\text{Regularizer}(\mathbf{Q}) = -\frac{1}{2} \sum_{i \in [b]} \log \left( \sum\nolimits_{m \in [M]} \alpha_{y_i,m}^{(t)} e^{-D_{KL,Lossy}\left(P_{U_i|x_i,w_e} \| Q_{y_i,m}^{(t)}\right)} \right)$$

$$-\frac{1}{2} \sum_{i \in [b]} \left( \frac{d}{2} \log\left(\pi e \sqrt{d}\right) + \log\left( \sum_{m=1}^{M} \alpha_{y_i,m}^{(t)} \tilde{t}_{i,m} \right) \right). \tag{59}$$

## D    FUTURE DIRECTIONS

In this work, we have established generalization bounds in terms of the minimum description length (MDL) of the latent variables. These bounds are particularly suitable for encoder-decoder architectures since they depend only on the encoder part of the model. The bounds improve the state-of-the-art results from $\sqrt{\text{MDL}(\mathbf{Q})/n}$ to $\text{MDL}(\mathbf{Q})/n$ in some cases.

Inspired by our established bounds, we propose a systematic approach to finding a data-dependent prior and using it as a regularizer. The approach consists of first finding the underlying "structure" of the latent variable space, modeling it as a Gaussian mixture and then steering the latent variables in

order to fit that mixture model. Conducted on various datasets and with various encoder architectures, reported experiments show promising results.

Our work opens up the door for several interesting future work directions, which we summarize hereafter.

1. In the main body of this work, we have established generalization bounds in terms of symmetric priors. However, the proposed practical approach for the design of the prior slightly violates the symmetry condition. While it is not uncommon (sometimes preferred ?) to stretch the technical assumptions for practical designs a little, in Appendix B we resolve the tension by showing that small deviations from the required technical symmetry only yield a small penalty in the bound. In the context of this paper, this result could be made more precise by studying the exact deviation of the proposed Gaussian mixture prior from the required symmetry and the caused deterioration of the bound.

2. The introduced regularizer depends on the dimension of the latent variable, rather than on the dimension of the model or the input data, which are often much larger. This is a major advantage of our approach. In addition, our approach is relatively easy to implement. Nevertheless, similar to many other regularizers, this comes at the expense of some additional computational overhead. Possible means of reducing that overhead include: (i) using the regularizer only in the first $K$ epochs (which, generally, are the most critical (KMN$^+$16; ARS17)) and (ii) applying the regularizer in a suitable lower-dimensional space, e.g., after proper projection of the latent vector onto that space.

3. In Section 4, we have shown how a weighted attention mechanism emerges naturally in the process of finding the data-dependent Gaussian mixture prior. This may be particularly interesting; and is worth further exploration especially when our approach is applied to self-attention layers.

4. In Section 4, proper selection of the number of components of the Gaussian mixture ($M$) should depend, among other factors, on the dimension $d$ of the problem and the number of hidden "subpopulations" in the latent vector (which itself depends on the used encoder!). Thus, suitable values of $M$ seem difficult to obtain beforehand; and, instead, one can resort to simply treating it as a hyper-parameter. One approach to circumventing this could be to explore the "structure" of the training data using some common dimensionality reduction and unsupervised clustering techniques, such as the t-SNE of (CRHC18) or the method of (YLL12).

5. Finally, we mention that in this work, we focused primarily on the application to classification tasks. However, the approach and results of this paper can be extended to other setups, such as semi-supervised and transfer learning settings.

## E  DETAILS OF THE EXPERIMENTS

This section provides additional details about the experiments that were conducted. The code used in the experiments is available at `https://github.com/PiotrKrasnowski/Gaussian_Mixture_Priors_for_Representation_Learning`.

### E.1  DATASETS

In all experiments, we used the following image classification datasets:

**CIFAR10** (KH$^+$09) - a dataset of 60,000 labeled images of dimension $32 \times 32 \times 3$ representing 10 different classes of animals and vehicles.

**CIFAR100** (KH$^+$09) - a dataset of 60,000 labeled images of dimension $32 \times 32 \times 3$ representing 100 different classes.

**USPS** (Hul94)[3] - a dataset of 9,298 labeled images of dimension $16 \times 16 \times 1$ representing 10 classes of handwritten digits.

---

[3]`https://www.csie.ntu.edu.tw/~cjlin/libsvmtools/datasets/multiclass.html#usps`

**INTEL**[4] - a dataset of over 24,000 labeled images of dimension $150 \times 150 \times 3$ representing 6 classes of different landscapes ('buildings', 'forest', 'glacier', 'mountain', 'sea', 'street').

All images were normalized before feeding them to the encoder.

### E.2 ARCHITECTURE DETAILS

The experiments were conducted using two types of encoder models: a custom convolutional encoder and a pre-trained ResNet18 followed by a linear layer (more specifically, the model "ResNet18_Weights.IMAGENET1K_V1" in PyTorch). The architecture of the CNN-based encoder can be found in Table 2. This custom encoder is a concatenation of four convolutional layers and two linear layers. We apply max-pooling and a LeakyReLU activation function with a negative slope coefficient set to $0.1$. The encoders take re-scaled images as input and generate parameters $\mu_x$ and variance $\sigma_x^2$ of the latent variable of dimension $m = 64$. Latent samples are produced using the reparameterization trick introduced by (KW14). Subsequently, the generated latent samples are fed into a decoder with a single linear layer and softmax activation function. The decoder's output is a soft class prediction.

Our tested encoders were complex enough to make them similar to "a universal function approximator", in line with (DKSV20). Conversely, we employ a straightforward decoder akin to (AFDM17) to minimize the unwanted regularization caused by a highly complex decoder. This approach allows us to emphasize the advantages of our regularizer in terms of generalization performance. However, note that the used ResNet18 model is already pre-trained using various regularization and data augmentation techniques. Therefore, the effect of a new regularizer is naturally less visible.

Table 2: The architecture of the convolutional encoder used in the experiments. The convolutional layers are parameterized respectively by the number of input channels, the number of output channels, and the filter size. The linear layers are defined by their input and output sizes.

| \multicolumn Encoder | | \multicolumn Encoder cont'd | | \multicolumn Encoder cont'd | |
|---|---|---|---|---|---|
| Number | Layer | Number | Layer | Number | Layer |
| 1 | Conv2D(3,8,5) | 6 | Conv2D(16,16,3) | 11 | LeakyReLU(0.1) |
| 2 | Conv2D(3,8,5) | 7 | LeakyReLU(0.1) | 12 | Linear(256,128) |
| 3 | LeakyReLU(0.1) | 8 | MaxPool(2,2) | \multicolumn Decoder | |
| 4 | MaxPool(2,2) | 9 | Flatten | 1 | Linear(64,10) |
| 5 | Conv2D(8,16,3) | 10 | Linear(N,256) | 2 | Softmax |

### E.3 IMPLEMENTATION AND TRAINING DETAILS

The PyTorch library (PGM[+]19) and a GPU Tesla P100 with CUDA 11.0 were utilized to train our prediction model. We employed the PyTorch Xavier initialization scheme (GB10) to initialize all weights, except biases set to zero. For optimization, we used the Adam optimizer (KB15) with parameters $\beta_1 = 0.5$ and $\beta_2 = 0.999$, an initial learning rate of $10^{-4}$, an exponential decay of 0.97, and a batch size of 128.

We trained the encoder and decoder models for 200 epochs five times independently for each considered regularization loss and for each value of the regularization parameter $\beta$ ranging between zero and one. The training was done using conventional cross-entropy loss for image category classification at the decoder's output, and regularization of the encoder's output based on either the standard VIB, the Category-dependent VIB, or our Gaussian mixture objective functions. For the Gaussian mixture objective function, we selected $M$=20 priors for each class category. The Gaussian mixture priors were initialized using the approaches in C.1. The priors were updated after each training iteration using the procedure in C.3 with a moving average coefficient $\eta_1 = 1\mathrm{e}-2$ for the priors' means $\mu_{c,m}$, $\eta_2 = 5\mathrm{e}-4$ for the priors' variances $\sigma_{c,m}^2$, and $\eta_3 = 1\mathrm{e}-2$ for the mixture weights $\alpha_{c,m}$. Following the approach outlined in (AFDM17), we generated one latent sample per image during training and 12 samples during testing.

---

[4]https://www.kaggle.com/datasets/puneet6060/intel-image-classification

## F KL-DIVERGENCE ESTIMATION

In this section, we first recall the KL-divergence estimation of two Gaussian mixture distributions developed in (HO07; DTK12). Then, we adapt these approaches to the case where the KL-divergence estimation of a Gaussian distribution and a Gaussian mixture distribution is considered.

### F.1 KL-DIVERGENCE ESTIMATION OF TWO GAUSSIAN MIXTURE DISTRIBUTIONS

In this section, we recall the results of (HO07; DTK12). We give the results only for the case where the covariance matrices of the Gaussian components are diagonal, for simplicity and because only diagonal covariance matrices are considered in our work. However, the results hold for the general form of the covariance matrix.

Consider two Gaussian mixture distributions $P$ and $Q$, defined as

$$P = \sum_{j=1}^{N} \beta_j P_j,$$

$$Q = \sum_{i=1}^{M} \alpha_i Q_i,$$

where $\alpha_i, \beta_j \geqslant 0$, $\sum_{j \in [N]} \beta_j = 1$, and $\sum_{i \in [M]} \alpha_i = 1$. In addition, each component is a multivariate Gaussian distribution with diagonal covariance matrices.

$$P_j = \mathcal{N}(\boldsymbol{\mu}_{p,j}, \mathrm{diag}(\boldsymbol{\sigma}_{p,j}^2)),$$
$$Q_i = \mathcal{N}(\boldsymbol{\mu}_{q,i}, \mathrm{diag}(\boldsymbol{\sigma}_{q,i}^2)).$$

#### F.1.1 PRODUCT OF GAUSSIAN APPROXIMATION

In this approximation, $D_{KL}(P\|Q)$ is approximated as (HO07):

$$D_{\mathrm{prod}}(P\|Q) := \sum_{j \in [N]} \beta_j \log \left( \frac{\sum_{j' \in [M]} \beta_{j'} \mathbb{E}_{P_j}[P_{j'}]}{\sum_{i \in [M]} \alpha_i \mathbb{E}_{P_j}[Q_i]} \right). \tag{60}$$

Note that this approximation is generally neither an upper bound nor a lower bound.

#### F.1.2 VARIATIONAL APPROXIMATION

In this approximation, $D_{KL}(P\|Q)$ is approximated as (HO07):

$$D_{\mathrm{var}}(P\|Q) := \sum_{j \in [N]} \beta_j \log \left( \frac{\sum_{j' \in [M]} \beta_{j'} e^{-D_{KL}(P_j\|P_{j'})}}{\sum_{i \in [M]} \alpha_i e^{-D_{KL}(P_j\|Q_i)}} \right). \tag{61}$$

Note that this approximation is again not an upper or lower bound in general.

#### F.1.3 AVERAGE OF TWO APPROXIMATIONS

It has been shown in (HO07; DTK12), that the average of the product and variational approximation provides a better estimate of the KL-divergence between two Gaussian prior distributions.

$$D_{\mathrm{est}}(P\|Q) = \frac{D_{\mathrm{prod}}(P\|Q) + D_{\mathrm{var}}(P\|Q)}{2}. \tag{62}$$

### F.2 KL-DIVERGENCE ESTIMATION BETWEEN A GAUSSIAN AND A GAUSSIAN MIXTURE DISTRIBUTION

In this section, we adapt the approaches of (HO07) for the setup where $P$ is a $d$-dimensional Gaussian distribution with a diagonal covariance matrix and $Q$ is a Gaussian mixture of $M$ of $d$-dimensional Gaussians with a diagonal covariance matrix.

Formally, let

$$P = \mathcal{N}(\boldsymbol{\mu}, \mathrm{diag}(\boldsymbol{\sigma}_p^2)),$$

and $Q$ be a Gaussian mixture

$$Q = \sum_{i=1}^{M} \alpha_i Q_i,$$

where $\alpha_i \geq 0$, $\sum_{i \in [M]} \alpha_i = 1$, and

$$Q_i = \mathcal{N}(\boldsymbol{\mu}_i, \mathrm{diag}(\boldsymbol{\sigma}_{q,i}^2)).$$

### F.2.1 PRODUCT OF GAUSSIAN BOUND

Denoting $L_P(f) := \mathbb{E}_P[\log(f)]$, we have $D_{KL}(P\|Q) = L_P(P) - L_P(Q)$. Note that

$$L_P(P) = -h(P) = -\frac{1}{2} \log\Big((2\pi e)^d \prod_{j \in [d]} \sigma_{p,j}^2\Big),$$

where $h(\cdot)$ is the differential entropy. Next, to bound $L_P(Q)$, using the idea of (HO07), we have

$$L_P(Q) = \mathbb{E}_P\Big[\log\big(\sum_{i=1}^{M} \alpha_i Q_i\big)\Big] \leq \log\Big(\sum_{i=1}^{M} \alpha_i \mathbb{E}_P[Q_i]\Big) = \log\Big(\sum_{i=1}^{M} \alpha_i t_i\Big), \tag{63}$$

where

$$t_i = \mathbb{E}_P[Q_i] = \int_x P(x)Q_i(x)\mathrm{d}x, \tag{64}$$

is the normalization constant of the product of the Gaussians (refer to (DTK12, Appendix A)). Note that by choice of the diagonal covariance matrices, these constants can be written as the product of $m$ coordinate-wise constants.

Thus, we have

$$D_{KL}(P\|Q) \geq -\frac{1}{2} \log\Big((2\pi e)^d \prod_{j \in [d]} \sigma_{p,j}^2\Big) - \log\Big(\sum_{i=1}^{M} \alpha_i t_i\Big) =: D_{\mathrm{prod}}(P\|Q). \tag{65}$$

Note that, unlike the KL divergence estimation of two Gaussian mixture priors, here the product of Gaussian approaches provides a lower bound.

### F.2.2 VARIATIONAL BOUND

Fix some $\gamma_i \geq 0$, $i \in [M]$ such that $\sum_i \gamma_i = 1$. Then,

$$
\begin{aligned}
L_P(Q) &= \mathbb{E}_P\Big[\log\big(\sum_{i=1}^{M} \alpha_i Q_i\big)\Big] \\
&= \mathbb{E}_P\Big[\log\big(\sum_{i=1}^{M} \alpha_i \gamma_i \frac{Q_i}{\gamma_i}\big)\Big] \\
&\geq \sum_{i \in [M]} \gamma_i \mathbb{E}_P\Big[\log\big(\frac{\alpha_i Q_i}{\gamma_i}\big)\Big] \\
&= \sum_{i \in [M]} \gamma_i\big(L_P(Q_i) + \log(\alpha_i/\gamma_i)\big).
\end{aligned}
\tag{66}
$$

Maximizing this lower bound with respect to $\gamma_i$ gives

$$\gamma_i = \frac{\alpha_i e^{-D_{KL}(P\|Q_i)}}{\sum_{i' \in [M]} \alpha_{i'} e^{-D_{KL}(P\|Q_{i'})}}. \tag{67}$$

Using this choice, (66) simplifies as

$$L_P(Q) = \mathbb{E}_P\Big[\log\big(\sum_{i=1}^{M}\alpha_i Q_i\big)\Big] \geqslant \log\bigg(\sum_{i\in[M]}\alpha_i e^{L_P(Q_i)}\bigg). \tag{68}$$

Hence, overall

$$D_{KL}(P\|Q) \leqslant L_P(P) - \log\bigg(\sum_{i\in[M]}\alpha_i e^{L_P(Q_i)}\bigg) \tag{69}$$

$$= -\log\bigg(\sum_{i\in[M]}\alpha_i e^{-D_{KL}(P\|Q_i)}\bigg) \tag{70}$$

$$=: D_{\mathrm{var}}(P\|Q). \tag{71}$$

Note that again, unlike the KL divergence estimation of two Gaussian mixture priors, where the variation approach provides only an approximation, this approach provides an upper bound.

### F.2.3 AVERAGE OF TWO APPROXIMATIONS

Finally, to estimate the KL-divergence between a Gaussian distribution and a Gaussian mixture distribution, we consider the average of the product of the Gaussian lower bound and the variational upper bound.

$$D_{\mathrm{est}}(P\|Q) = \frac{D_{\mathrm{prod}}(P\|Q) + D_{\mathrm{var}}(P\|Q)}{2}. \tag{72}$$

## G PROOFS

In this section, we present the deferred proofs.

### G.1 PROOF OF THEOREM 1

Fix some symmetric conditional prior $\mathbf{Q}(\mathbf{U}, \mathbf{U}'|\mathbf{Y}, \mathbf{Y}', \mathbf{X}, \mathbf{X}', W_e)$. We will show that

$$\mathbb{E}_{\mathbf{S},\mathbf{S}',W,\hat{\mathbf{Y}},\hat{\mathbf{Y}}'}\left[h_D\Big(\hat{\mathcal{L}}(\mathbf{Y}', \hat{\mathbf{Y}}'), \hat{\mathcal{L}}(\mathbf{Y}, \hat{\mathbf{Y}})\Big) - h_{\mathbf{Y},\mathbf{Y}',\hat{\mathbf{Y}},\hat{\mathbf{Y}}'}\Big(\frac{1}{2}\|\hat{p}_{\mathbf{Y}} - \hat{p}_{\mathbf{Y}'}\|_1\Big)\right] \leqslant \frac{\mathrm{MDL}(\mathbf{Q}) + \log(n)}{n}, \tag{73}$$

where $\hat{p}_{\mathbf{Y}}$ and $\hat{p}_{\mathbf{Y}'}$ are empirical distributions of $\mathbf{Y}$ and $\mathbf{Y}'$, respectively,

$$\mathrm{MDL}(\mathbf{Q}) := \mathbb{E}_{S,S',W_e}\Big[D_{KL}\Big(P_{\mathbf{U},\mathbf{U}'|\mathbf{X},\mathbf{X}',W_e}\Big\|\mathbf{Q}\Big)\Big], \tag{74}$$

and

$$(\mathbf{S}, \mathbf{S}', \mathbf{U}, \mathbf{U}', \hat{\mathbf{Y}}, \hat{\mathbf{Y}}', W) \sim P_{S,W}P_{S'}P_{\mathbf{U}|\mathbf{X},W_e}P_{\mathbf{U}'|\mathbf{X}',W_e}P_{\hat{\mathbf{Y}}|\mathbf{U},W_d}P_{\hat{\mathbf{Y}}'|\mathbf{U}',W_d}.$$

Denote

$$P_1 := P_{S,W}P_{S'}P_{\mathbf{U}|\mathbf{X},W_e}P_{\mathbf{U}'|\mathbf{X}',W_e}P_{\hat{\mathbf{Y}}|\mathbf{U},W_d}P_{\hat{\mathbf{Y}}'|\mathbf{U}',W_d},$$

$$P_2 := P_{S,W}P_{S'}Q_{\mathbf{U},\mathbf{U}'|\mathbf{X},\mathbf{X}',\mathbf{Y},\mathbf{Y}',W_e}P_{\hat{\mathbf{Y}}|\mathbf{U},W_d}P_{\hat{\mathbf{Y}}'|\mathbf{U}',W_d},$$

$$f\Big(\mathbf{Y}, \mathbf{Y}', \hat{\mathbf{Y}}, \hat{\mathbf{Y}}'\Big) := h_D\Big(\hat{\mathcal{L}}(\mathbf{Y}', \hat{\mathbf{Y}}'), \hat{\mathcal{L}}(\mathbf{Y}, \hat{\mathbf{Y}})\Big) - h_{\mathbf{Y},\mathbf{Y}',\hat{\mathbf{Y}},\hat{\mathbf{Y}}'}\Big(\frac{1}{2}\|\hat{p}_{\mathbf{Y}} - \hat{p}_{\mathbf{Y}'}\|_1\Big).$$

Next, similar to information-theoretic (e.g. (XR17; SZ20; SZK23)) and PAC-Bayes-based approaches (e.g. (Alq21; RKSST20)) we use Donsker-Varadhan's inequality to change the measure from $P_1$ to $P_2$. The cost of such a change is $D_{KL}(P_1\|P_2) = \mathrm{MDL}(\mathbf{Q})$. We apply Donsker-Varadhan on the function $nf$. Concretely, we have

$$\mathbb{E}_{\mathbf{S},\mathbf{S}',W,\hat{\mathbf{Y}},\hat{\mathbf{Y}}'}\Big[f\Big(\mathbf{Y}, \mathbf{Y}', \hat{\mathbf{Y}}, \hat{\mathbf{Y}}'\Big)\Big] \leqslant D_{KL}(P_1\|P_2) + \log\Big(\mathbb{E}_{P_2}\Big[e^{nf(\mathbf{Y},\mathbf{Y}',\hat{\mathbf{Y}},\hat{\mathbf{Y}}')}\Big]\Big)$$

$$= \mathrm{MDL}(\mathbf{Q}) + \log\Big(\mathbb{E}_{P_2}\Big[e^{nf(\mathbf{Y},\mathbf{Y}',\hat{\mathbf{Y}},\hat{\mathbf{Y}}')}\Big]\Big).$$

Hence, it remains to show that

$$\mathbb{E}_{P_2}\left[e^{nf(\mathbf{Y},\mathbf{Y}',\hat{\mathbf{Y}},\hat{\mathbf{Y}}')}\right] \leqslant n. \tag{75}$$

Let $\tilde{\mathbf{Q}}_{\hat{\mathbf{Y}},\hat{\mathbf{Y}}'|\mathbf{Y},\mathbf{Y}'}$ be the conditional distribution of $(\hat{\mathbf{Y}},\hat{\mathbf{Y}}')$ given $(\mathbf{Y},\mathbf{Y}')$, under the joint distribution $P_2$. It can be easily verified that $\tilde{\mathbf{Q}}$ satisfies the symmetry property since $\mathbf{Q}$ is symmetric (as defined in Definition 1). For better clarity, we re-state the symmetry property of $\tilde{\mathbf{Q}}$ and define some notations that will be used in the rest of the proof.

Let $Y^{2n} := (\mathbf{Y},\mathbf{Y}')$ and $\hat{Y}^{2n} := (\hat{\mathbf{Y}},\hat{\mathbf{Y}}')$. For a given permutation $\tilde{\pi}: [2n] \rightarrow [2n]$, the permuted vectors $Y_{\tilde{\pi}}^{2n}$ and $\hat{Y}_{\tilde{\pi}}^{2n}$ are defined as

$$\begin{aligned} Y_{\tilde{\pi}}^{2n} &:= Y_{\tilde{\pi}(1)},\ldots,Y_{\tilde{\pi}(2n)}, \\ \hat{Y}_{\tilde{\pi}}^{2n} &:= \hat{Y}_{\tilde{\pi}(1)},\ldots,\hat{Y}_{\tilde{\pi}(2n)}. \end{aligned} \tag{76}$$

Furthermore, under the permutation $\tilde{\pi}$, we denote the first $n$ coordinates of $Y_{\tilde{\pi}}^{2n}$ and $\hat{Y}_{\tilde{\pi}}^{2n}$ by

$$\begin{aligned} \mathbf{Y}_{\tilde{\pi}} &:= Y_{\tilde{\pi}(1)},\ldots,\hat{Y}_{\tilde{\pi}(n)}, \\ \hat{\mathbf{Y}}_{\tilde{\pi}} &:= \hat{Y}_{\tilde{\pi}(1)},\ldots,\hat{Y}_{\tilde{\pi}(n)}, \end{aligned} \tag{77}$$

respectively, and the next $n$ coordinates of $Y_{\tilde{\pi}}^{2n}$ and $\hat{Y}_{\tilde{\pi}}^{2n}$ by

$$\begin{aligned} \mathbf{Y}_{\tilde{\pi}}' &:= Y_{\tilde{\pi}(n+1)},\ldots,Y_{\tilde{\pi}(2n)}, \\ \hat{\mathbf{Y}}_{\tilde{\pi}}' &:= \hat{Y}_{\tilde{\pi}(n+1)},\ldots,\hat{Y}_{\tilde{\pi}(2n)}. \end{aligned} \tag{78}$$

respectively. By $\tilde{\mathbf{Q}}$ being symmetric, we mean that $\tilde{\mathbf{Q}}_{\hat{\mathbf{Y}}_{\tilde{\pi}},\hat{\mathbf{Y}}_{\tilde{\pi}}'|\mathbf{Y},\mathbf{Y}'}$ remains invariant under all permutations such that $Y_i = Y_{\tilde{\pi}(i)}$ for all $i \in [2n]$. In other words, all permutations such that $\mathbf{Y} = \mathbf{Y}_{\tilde{\pi}}$ and $\mathbf{Y}' = \mathbf{Y}_{\tilde{\pi}}'$.

Hence, we can write

$$\mathbb{E}_{P_2}\left[e^{nf(\mathbf{Y},\mathbf{Y}',\hat{\mathbf{Y}},\hat{\mathbf{Y}}')}\right] = \mathbb{E}_{\mathbf{Y},\mathbf{Y}',\hat{\mathbf{Y}},\hat{\mathbf{Y}}'}\left[e^{nf(\mathbf{Y},\mathbf{Y}',\hat{\mathbf{Y}},\hat{\mathbf{Y}}')}\right], \tag{79}$$

where $\mathbf{Y},\mathbf{Y}',\hat{\mathbf{Y}},\hat{\mathbf{Y}}' \sim \mu_Y^{\otimes 2n}\tilde{\mathbf{Q}}_{\hat{\mathbf{Y}},\hat{\mathbf{Y}}'|\mathbf{Y},\mathbf{Y}'}$.

Fix some $\mathbf{Y}$ and $\mathbf{Y}'$. Without loss of generality and for simplicity, assume that $\mathbf{Y}$ and $\mathbf{Y}'$ are *ordered*, in the sense that for $r \in [R]$, $Y_r = Y_r'$, and $\{Y_{R+1},\ldots,Y_n\} \bigcap \{Y_{R+1}',\ldots,Y_n'\} = \varnothing$, where

$$R = n - \frac{n}{2}\|\hat{p}_{\mathbf{Y}} - \hat{p}_{\mathbf{Y}'}\|_1.$$

Otherwise, it is easy to see that the following analysis holds by proper (potentially non-identical) re-orderings of $\mathbf{Y}$ and $\mathbf{Y}'$ and corresponding predictions $\hat{\mathbf{Y}}$ (according to the way $\mathbf{Y}$ is re-ordered) and $\hat{\mathbf{Y}}'$ (according to the way $\mathbf{Y}'$ is re-ordered), such that $\mathbf{Y}$ and $\mathbf{Y}'$ coincidence in all first $R$ coordinates and do not have any overlap in the remaining $n - R$ coordinates.

Furthermore, for $r \in [n]$, let $J_r \in \{r, n + r\} \sim \text{Bern}(\frac{1}{2})$ be a uniform binary random variable and define $J_r^c$ as its complement, *i.e.*, $J_r \cup J_r^c = \{r, n + r\}$. Define the mapping $\pi_R := [2n] \rightarrow [2n]$ as following: For $r \in [R]$, $\pi_R(r) = J_r$ and $\pi_R(r + n) = J_r^c$. For $r \in [R + 1, n]$, $\pi_R(r) = r$ and $\pi_R(n + r) = n + r$. Note that $\pi_R$ depends on $(\mathbf{Y},\mathbf{Y}')$ and under $\pi_R$, $\mathbf{Y} = \mathbf{Y}_{\pi_R}$ and $\mathbf{Y}' = \mathbf{Y}_{\pi_R}'$, where $\mathbf{Y}_{\pi_R}$ and $\mathbf{Y}_{\pi_R}'$ are defined in (77) and (78), respectively. Hence, $\|\hat{p}_{\mathbf{Y}} - \hat{p}_{\mathbf{Y}'}\|_1 = \left\|\hat{p}_{\mathbf{Y}_{\pi_R}} - \hat{p}_{\mathbf{Y}_{\pi_R}'}\right\|_1$. To simplify the notations, in what follows we denote the coordinates of $\mathbf{Y}_{\pi_R}$ by

$$\mathbf{Y}_{\pi_R} := (Y_{\pi_R,1},\ldots,Y_{\pi_R,n}),$$

and the coordinates of $\mathbf{Y}_{\pi_R}'$ by

$$\mathbf{Y}_{\pi_R}' := (Y_{\pi_R,1}',\ldots,Y_{\pi_R,n}').$$

Note that by (77) and (78), we have $Y_{\pi_R,i} = Y^{2n}_{\pi_R(i)}$ and $Y'_{\pi_R,i} = Y^{2n}_{\pi_R(i+n)}$ for $i \in [n]$, where $Y^{2n}_{\pi_R(i)}$ is defined in (76). Similar notations are used for the prediction vectors, *i.e.*,

$$\hat{\mathbf{Y}}_{\pi_R} := (\hat{Y}_{\pi_R,1}, \ldots, \hat{Y}_{\pi_R,n}),$$
$$\hat{\mathbf{Y}}'_{\pi_R} := (\hat{Y}'_{\pi_R,1}, \ldots, \hat{Y}'_{\pi_R,n}).$$

With these notations, for a fixed ordered $\mathbf{Y}$ and $\mathbf{Y}'$ we have

$$\mathbb{E}_{\hat{\mathbf{Y}},\hat{\mathbf{Y}}'|\mathbf{Y},\mathbf{Y}'}\left[e^{nf(\mathbf{Y},\mathbf{Y}',\hat{\mathbf{Y}},\hat{\mathbf{Y}}')}\right] = \mathbb{E}_{\hat{\mathbf{Y}},\hat{\mathbf{Y}}'|\mathbf{Y},\mathbf{Y}'}\mathbb{E}_{J_1,\ldots,J_R\sim\mathrm{Bern}(\frac{1}{2})^{\otimes R}}\left[e^{nf(\mathbf{Y},\mathbf{Y}',\hat{\mathbf{Y}}_{\pi_R},\hat{\mathbf{Y}}'_{\pi_R})}\right]$$
$$= \mathbb{E}_{\hat{\mathbf{Y}},\hat{\mathbf{Y}}'|\mathbf{Y},\mathbf{Y}'}\mathbb{E}_{J_1,\ldots,J_R\sim\mathrm{Bern}(\frac{1}{2})^{\otimes R}}\left[e^{nf(\mathbf{Y}_{\pi_R},\mathbf{Y}'_{\pi_R},\hat{\mathbf{Y}}_{\pi_R},\hat{\mathbf{Y}}'_{\pi_R})}\right]. \quad (80)$$

where the first step follows due to the symmetric property of $\tilde{\mathbf{Q}}$ and the second step follows since $\mathbf{Y} = \mathbf{Y}_{\pi_R}$ and $\mathbf{Y}' = \mathbf{Y}'_{\pi_R}$.

Now, consider another mapping $\pi := [2n] \to [2n]$ such that $\pi$ is identical to $\pi_R$ for the indices in the range $[1:R] \cup [n+1:n+R]$, *i.e.*, for $r \in [R]$,

$$\pi(r) = \pi_R(r) = J_r, \qquad \pi(r+n) = \pi_R(r+n) = J^c_r.$$

Furthermore, for the indices in the range in $[R+1:n] \cup [n+R+1:2n]$, $\pi$ is defined as follows: for $r \in [R+1, n]$,

$$\pi(r) = J_r, \qquad \pi(n+r) = J^c_r,$$

where as previously defined, $J_r \in \{r, n+r\} \sim \mathrm{Bern}(\frac{1}{2})$ is a uniform binary random variable and $J^c_r$ is its complement. Denote

$$J^n_{R+1} := J_{R+1}, \ldots, J_n.$$

With the above definitions, we have

$$e^{nf(\mathbf{Y}_{\pi_R},\mathbf{Y}'_{\pi_R},\hat{\mathbf{Y}}_{\pi_R},\hat{\mathbf{Y}}'_{\pi_R})}$$

$$= \mathbb{E}_{J^n_{R+1}\sim\mathrm{Bern}(\frac{1}{2})^{\otimes(n-R)}}\left[e^{nh_D\left(\frac{1}{n}\sum_{i=1}^n \mathbb{1}_{\{\hat{Y}'_{\pi,i}\neq Y'_{\pi,i}\}}, \frac{1}{n}\sum_{i=1}^n \mathbb{1}_{\{\hat{Y}_{\pi,i}\neq Y_{\pi,i}\}}\right)}\right.$$

$$\left. \times e^{nf(\mathbf{Y}_{\pi_R},\mathbf{Y}'_{\pi_R},\hat{\mathbf{Y}}_{\pi_R},\hat{\mathbf{Y}}'_{\pi_R})-nh_D\left(\frac{1}{n}\sum_{i=1}^n \mathbb{1}_{\{\hat{Y}'_{\pi,i}\neq Y'_{\pi,i}\}}, \frac{1}{n}\sum_{i=1}^n \mathbb{1}_{\{\hat{Y}_{\pi,i}\neq Y_{\pi,i}\}}\right)}\right]$$

$$\overset{(a)}{\leqslant} \mathbb{E}_{J^n_{R+1}\sim\mathrm{Bern}(\frac{1}{2})^{\otimes(n-R)}}\left[e^{nh_D\left(\frac{1}{n}\sum_{i=1}^n \mathbb{1}_{\{\hat{Y}'_{\pi,i}\neq Y'_{\pi,i}\}}, \frac{1}{n}\sum_{i=1}^n \mathbb{1}_{\{\hat{Y}_{\pi,i}\neq Y_{\pi,i}\}}\right)}\right], \quad (81)$$

where $(a)$ holds due to the following Lemma, shown in Appendix G.5.

**Lemma 1.** *The below relation holds:*

$$f\left(\mathbf{Y}_{\pi_R}, \mathbf{Y}'_{\pi_R}, \hat{\mathbf{Y}}_{\pi_R}, \hat{\mathbf{Y}}'_{\pi_R}\right) \leqslant h_D\left(\frac{1}{n}\sum_{i=1}^n \mathbb{1}_{\{\hat{Y}'_{\pi,i}\neq Y'_{\pi,i}\}}, \frac{1}{n}\sum_{i=1}^n \mathbb{1}_{\{\hat{Y}_{\pi,i}\neq Y_{\pi,i}\}}\right). \quad (82)$$

Hence, for a fixed ordered $\mathbf{Y}$ and $\mathbf{Y}'$, combining (80) and (81) yields

$$\mathbb{E}_{\hat{\mathbf{Y}},\hat{\mathbf{Y}}'|\mathbf{Y},\mathbf{Y}'}\left[e^{nf(\mathbf{Y},\mathbf{Y}',\hat{\mathbf{Y}},\hat{\mathbf{Y}}')}\right]$$

$$= \mathbb{E}_{\hat{\mathbf{Y}},\hat{\mathbf{Y}}'|\mathbf{Y},\mathbf{Y}'}\mathbb{E}_{J_1,\ldots,J_n\sim\mathrm{Bern}(\frac{1}{2})^{\otimes n}}\left[e^{nh_D\left(\frac{1}{n}\sum_{i=1}^n \mathbb{1}_{\{\hat{Y}'_{\pi,i}\neq Y'_{\pi,i}\}}, \frac{1}{n}\sum_{i=1}^n \mathbb{1}_{\{\hat{Y}_{\pi,i}\neq Y_{\pi,i}\}}\right)}\right]$$

$$\leqslant n, \quad (83)$$

where the last step is derived by using (SZK23, Proof of Theorme 3). As mentioned before, it is easy to see that the above analysis holds for non-ordered $\mathbf{Y}$ and $\mathbf{Y}'$, by simply considering proper (potentially non-identical) re-orderings of $\mathbf{Y}$ and $\mathbf{Y}'$ and corresponding predictions $\hat{\mathbf{Y}}$ (according to the way $\mathbf{Y}$ is re-ordered) and $\hat{\mathbf{Y}}'$ (according to the way $\mathbf{Y}'$ is re-ordered), such that $\mathbf{Y}$ and $\mathbf{Y}'$ coincidence in all first $R$ coordinates and do not have any overlap in the remaining $n - R$ coordinates.

Combining (79), (80), and (83), shows (75) which completes the proof.

## G.2 PROOF OF THEOREM 2

First note that by convexity of the function $h_D$ ((SZK23, Lemma 1)), we have

$$h_D\Big(\hat{\mathcal{L}}(S',W),\hat{\mathcal{L}}(S,W)\Big) \leqslant \mathbb{E}_{\hat{\mathbf{Y}},\hat{\mathbf{Y}}'|\mathbf{Y},\mathbf{Y}'}\Big[h_D\Big(\hat{\mathcal{L}}(\mathbf{Y}',\hat{\mathbf{Y}}'),\hat{\mathcal{L}}(\mathbf{Y},\hat{\mathbf{Y}})\Big)\Big]. \tag{84}$$

Hence, it suffices to show that with probability at least $1-\delta$ over choices of $(S,S',W)$,

$$\mathbb{E}_{\hat{\mathbf{Y}},\hat{\mathbf{Y}}'|\mathbf{Y},\mathbf{Y}'}\Big[h_D\Big(\hat{\mathcal{L}}(\mathbf{Y}',\hat{\mathbf{Y}}'),\hat{\mathcal{L}}(\mathbf{Y},\hat{\mathbf{Y}})\Big)\Big] \leqslant \frac{D_{KL}\big(P_{\mathbf{U},\mathbf{U}'|\mathbf{X},\mathbf{X}',W_e}\big\|\mathbf{Q}\big) + \log(n/\delta)}{n}$$
$$+\mathbb{E}_{\hat{\mathbf{Y}},\hat{\mathbf{Y}}'|\mathbf{Y},\mathbf{Y}'}\Big[h_{\mathbf{Y},\mathbf{Y}',\hat{\mathbf{Y}},\hat{\mathbf{Y}}'}\Big(\frac{1}{2}\|\hat{p}_{\mathbf{Y}}-\hat{p}_{\mathbf{Y}'}\|_1\Big)\Big]. \tag{85}$$

Similar to the proof of Theorem 1, define

$$P_1' := P_{\mathbf{U}|\mathbf{X},W_e}P_{\mathbf{U}'|\mathbf{X}',W_e}P_{\hat{\mathbf{Y}}|\mathbf{U},W_d}P_{\hat{\mathbf{Y}}'|\mathbf{U}',W_d},$$
$$P_2' := Q_{\mathbf{U},\mathbf{U}'|\mathbf{X},\mathbf{X}',\mathbf{Y},\mathbf{Y}',W_e}P_{\hat{\mathbf{Y}}|\mathbf{U},W_d}P_{\hat{\mathbf{Y}}'|\mathbf{U}',W_d},$$
$$f\Big(\mathbf{Y},\mathbf{Y}',\hat{\mathbf{Y}},\hat{\mathbf{Y}}'\Big) := h_D\Big(\hat{\mathcal{L}}(\mathbf{Y}',\hat{\mathbf{Y}}'),\hat{\mathcal{L}}(\mathbf{Y},\hat{\mathbf{Y}})\Big) - h_{\mathbf{Y},\mathbf{Y}',\hat{\mathbf{Y}},\hat{\mathbf{Y}}'}\Big(\frac{1}{2}\|\hat{p}_{\mathbf{Y}}-\hat{p}_{\mathbf{Y}'}\|_1\Big).$$

Using Donsker-Varadhan's inequality, we have

$$n\mathbb{E}_{\hat{\mathbf{Y}},\hat{\mathbf{Y}}'|\mathbf{Y},\mathbf{Y}'}\Big[f\Big(\mathbf{Y},\mathbf{Y}',\hat{\mathbf{Y}},\hat{\mathbf{Y}}'\Big)\Big] \leqslant D_{KL}\big(P_1'\|P_2'\big) + \log\Big(\mathbb{E}_{P_2'}\Big[e^{nf\big(\mathbf{Y},\mathbf{Y}',\hat{\mathbf{Y}},\hat{\mathbf{Y}}'\big)}\Big]\Big)$$
$$= D_{KL}\big(P_{\mathbf{U},\mathbf{U}'|\mathbf{X},\mathbf{X}',W_e}\big\|\mathbf{Q}\big) + \log\Big(\mathbb{E}_{P_2'}\Big[e^{nf\big(\mathbf{Y},\mathbf{Y}',\hat{\mathbf{Y}},\hat{\mathbf{Y}}'\big)}\Big]\Big). \tag{86}$$

Hence,

$$\mathbb{P}\Bigg(\mathbb{E}_{\hat{\mathbf{Y}},\hat{\mathbf{Y}}'|\mathbf{Y},\mathbf{Y}'}\Big[f\Big(\mathbf{Y},\mathbf{Y}',\hat{\mathbf{Y}},\hat{\mathbf{Y}}'\Big)\Big] > \frac{D_{KL}\big(P_{\mathbf{U},\mathbf{U}'|\mathbf{X},\mathbf{X}',W_e}\big\|\mathbf{Q}\big) + \log(n/\delta)}{n}\Bigg)$$
$$\overset{(a)}{\leqslant} \mathbb{P}\Big(\log\Big(\mathbb{E}_{P_2'}\Big[e^{nf\big(\mathbf{Y},\mathbf{Y}',\hat{\mathbf{Y}},\hat{\mathbf{Y}}'\big)}\Big]\Big) > \log(n/\delta)\Big)$$
$$= \mathbb{P}\Big(\mathbb{E}_{P_2'}\Big[e^{nf\big(\mathbf{Y},\mathbf{Y}',\hat{\mathbf{Y}},\hat{\mathbf{Y}}'\big)}\Big] > n/\delta\Big)$$
$$\overset{(b)}{\leqslant} \frac{\mathbb{E}_{S,S',W_e}\mathbb{E}_{P_2'}\Big[e^{nf\big(\mathbf{Y},\mathbf{Y}',\hat{\mathbf{Y}},\hat{\mathbf{Y}}'\big)}\Big]}{n/\delta}$$
$$\overset{(c)}{\leqslant} \delta, \tag{87}$$

where

- $(a)$ follows by (86),
- $(b)$ is derived using the Markov inequality,
- and $(c)$ is shown in (75).

This completes the proof.

## G.3 PROOF OF PROPOSITION 1

To state the proof, first, we need to recall the notion of $\beta$-approximate max-information; as previously defined in (DR18, Definition 3.2) and (DR18, Definition 3.2). Here, we state the definition adapted to our setup. For ease of notation, denote

$$e_V(S) := (S,\mathcal{A}(S)) = (S,g(S,V)). \tag{88}$$

**Definition 5.** *Let $\beta \geqslant 0$. Then, define the $\beta$-max-information between $S$ and $\mathbf{Q}^{e_V(S)}$, denoted by $I_\infty^\beta$, as the minimal value $k$ such that for all product events $E$ and all fixed $V$, we have*

$$\mathbb{P}\Big(\big(S,\mathbf{Q}^{e_V(S)}\big) \in E\Big) \leqslant e^k\mathbb{P}\Big(\big(S,\mathbf{Q}^{e_V(\tilde{S})}\big) \in E\Big) + \beta, \tag{89}$$

*where $\tilde{S}$ is an independent dataset with the same distribution as $S$.*

Fix some $\delta' > 0$, which will be made explicit in the following. Now, "similar" to the proof of (DR18, Theorem 4.2), for any $\mathbf{Q} \in \mathcal{Q}$, define

$$R(\mathbf{Q}) = \left\{ (S, S', W) \colon h_D\left(\hat{\mathcal{L}}(S', W), \hat{\mathcal{L}}(S, W)\right) > \Delta(S, S', W, \mathbf{Q}, \delta') \right\}, \tag{90}$$

where

$$\Delta(S, S', W, \mathbf{Q}, \delta') := \frac{D_{KL}\left(P_{\mathbf{U}, \mathbf{U}'|\mathbf{X}, \mathbf{X}', W_e} \| \mathbf{Q}\right) + \log(n/\delta')}{n}$$
$$+ \mathbb{E}_{\hat{\mathbf{Y}}, \hat{\mathbf{Y}}'|\mathbf{Y}, \mathbf{Y}'}\left[ h_{\mathbf{Y}, \mathbf{Y}', \hat{\mathbf{Y}}, \hat{\mathbf{Y}}'}\left(\frac{1}{2}\|\hat{p}_{\mathbf{Y}} - \hat{p}_{\mathbf{Y}'}\|_1\right)\right]. \tag{91}$$

Fix some $\beta > 0$. For every fixed $S'$ and $V$, by Definition 5, we know that

$$\mathbb{P}\left((S, W, S') \in R(\mathbf{Q}^{e(S)})|S', V\right) \leqslant e^{I_\infty^\beta} \mathbb{P}\left((S, W, S') \in R(\mathbf{Q}^{ev(\tilde{S})})|S', V\right) + \beta,$$

where $\tilde{S}$ is independent of $(e(S), S')$. Hence,

$$\mathbb{P}_{S, W, S'}\left((S, W, S') \in R(\mathbf{Q}^{ev(S)})\right) \leqslant e^{I_\infty^\beta} \mathbb{P}_{S, W, S'}\left((S, W, S') \in R(\mathbf{Q}^{ev(\tilde{S})})\right) + \beta$$
$$\overset{(a)}{\leqslant} e^{I_\infty^\beta} \delta' + \beta, \tag{92}$$

where $(a)$ is derived since by Theorem 2, we know that $\mathbb{P}(R(\mathbf{Q})) \leqslant \delta'$ for every $\mathbf{Q}$ independent of $S$ and $S'$. Recall that strong symmetry implies symmetry.

Let $\beta = \delta/2$ and $\delta := e^{I_\infty^{\delta/2}} \delta' + \delta/2$. Equivalently,

$$\delta' := \frac{\delta e^{-I_\infty^{\delta/2}}}{2}.$$

With these choices, with probability $1 - \delta$ over choices of $(S, S', W)$, we have

$$h_D\left(\hat{\mathcal{L}}(S', W), \hat{\mathcal{L}}(S, W)\right) \leqslant \Delta(S, S', W, \mathbf{Q}, \delta')$$
$$= \frac{D_{KL}\left(P_{\mathbf{U}, \mathbf{U}'|\mathbf{X}, \mathbf{X}', W_e} \| \mathbf{Q}^{e(S)}\right) + \log(2n/\delta) + I_\infty^{\delta/2}}{n}$$
$$+ \mathbb{E}_{\hat{\mathbf{Y}}, \hat{\mathbf{Y}}'|\mathbf{Y}, \mathbf{Y}'}\left[ h_{\mathbf{Y}, \mathbf{Y}', \hat{\mathbf{Y}}, \hat{\mathbf{Y}}'}\left(\frac{1}{2}\|\hat{p}_{\mathbf{Y}} - \hat{p}_{\mathbf{Y}'}\|_1\right)\right]. \tag{93}$$

The final result follows by (DFH$^+$15, Theorem 20), where they showed that

$$I_\infty^{\delta/2} \leqslant \frac{n}{2}\varepsilon_p^2 + \varepsilon_p \sqrt{\frac{n \log(4/\delta)}{2}}.$$

This completes the proof.

## G.4 PROOF OF PROPOSITION 2

Recall the following notations in the proof of Theorem 1:

$$P_1 := P_{S, W} P_{S'} P_{\mathbf{U}|\mathbf{X}, W_e} P_{\mathbf{U}'|\mathbf{X}', W_e} P_{\hat{\mathbf{Y}}|\mathbf{U}, W_d} P_{\hat{\mathbf{Y}}'|\mathbf{U}', W_d},$$
$$P_2 := P_{S, W} P_{S'} Q_{\mathbf{U}, \mathbf{U}'|\mathbf{X}, \mathbf{X}', \mathbf{Y}, \mathbf{Y}', W_e} P_{\hat{\mathbf{Y}}|\mathbf{U}, W_d} P_{\hat{\mathbf{Y}}'|\mathbf{U}', W_d},$$
$$f\left(\mathbf{Y}, \mathbf{Y}', \hat{\mathbf{Y}}, \hat{\mathbf{Y}}'\right) := h_D\left(\hat{\mathcal{L}}(\mathbf{Y}', \hat{\mathbf{Y}}'), \hat{\mathcal{L}}(\mathbf{Y}, \hat{\mathbf{Y}})\right) - h_{\mathbf{Y}, \mathbf{Y}', \hat{\mathbf{Y}}, \hat{\mathbf{Y}}'}\left(\frac{1}{2}\|\hat{p}_{\mathbf{Y}} - \hat{p}_{\mathbf{Y}'}\|_1\right).$$

Using the identical steps as in the proof Theorem 1, we have

$$\mathbb{E}_{\mathbf{S}, \mathbf{S}', W, \hat{\mathbf{Y}}, \hat{\mathbf{Y}}'}\left[f\left(\mathbf{Y}, \mathbf{Y}', \hat{\mathbf{Y}}, \hat{\mathbf{Y}}'\right)\right] \leqslant \mathrm{MDL}(\mathbf{Q}) + \log\left(\mathbb{E}_{P_2}\left[e^{nf\left(\mathbf{Y}, \mathbf{Y}', \hat{\mathbf{Y}}, \hat{\mathbf{Y}}'\right)}\right]\right).$$

Hence, it remains to show that

$$\mathbb{E}_{P_2}\left[e^{nf\left(\mathbf{Y}, \mathbf{Y}', \hat{\mathbf{Y}}, \hat{\mathbf{Y}}'\right)}\right] \leqslant \delta e^{2n} + n e^\epsilon. \tag{94}$$

Let $\Pi_{\mathbf{Y},\mathbf{Y}'}$ denote the set of all permutations that preserve the labeling. Denote the size of this set as $N_{\pi,\mathbf{Y},\mathbf{Y}'} := N$. Then, the prior

$$\tilde{\mathbf{Q}}(\mathbf{U}, \mathbf{U}'|\mathbf{Y}, \mathbf{Y}', \mathbf{X}, \mathbf{X}', W_e) := \frac{1}{N} \sum_{\pi \in \Pi_{\mathbf{Y},\mathbf{Y}'}} \mathbf{Q}(\mathbf{U}_\pi, \mathbf{U}'_\pi|\mathbf{Y}, \mathbf{Y}', \mathbf{X}, \mathbf{X}', W_e), \tag{95}$$

is symmetric in the sense of Definition 1. Furthermore, by Definition 4, we have with probability at least $1 - \delta$ over choices of $(S', S, W_e, \mathbf{U}, \mathbf{U}') \sim P_{S'} P_{S,W_e} \mathbf{Q}$,

$$\mathbf{Q}(\mathbf{U}, \mathbf{U}'|\mathbf{Y}, \mathbf{Y}', \mathbf{X}, \mathbf{X}', W_e) \leqslant e^\epsilon \tilde{\mathbf{Q}}(\mathbf{U}, \mathbf{U}'|\mathbf{Y}, \mathbf{Y}', \mathbf{X}, \mathbf{X}', W_e). \tag{96}$$

Hence, since $f\left(\mathbf{Y}, \mathbf{Y}', \hat{\mathbf{Y}}, \hat{\mathbf{Y}}'\right) \leqslant 2$, we have that

$$\mathbb{E}_{P_2}\left[e^{nf\left(\mathbf{Y},\mathbf{Y}',\hat{\mathbf{Y}},\hat{\mathbf{Y}}'\right)}\right] \leqslant \delta e^{2n} + e^\epsilon \mathbb{E}_{P_3}\left[e^{nf\left(\mathbf{Y},\mathbf{Y}',\hat{\mathbf{Y}},\hat{\mathbf{Y}}'\right)}\right], \tag{97}$$

where

$$P_3 := P_{S,W} P_{S'} \tilde{Q}_{\mathbf{U},\mathbf{U}'|\mathbf{X},\mathbf{X}',\mathbf{Y},\mathbf{Y}',W_e} P_{\hat{\mathbf{Y}}|\mathbf{U},W_d} P_{\hat{\mathbf{Y}}'|\mathbf{U}',W_d}.$$

The result now follows since $\tilde{Q}$ is symmetric and hence identical to the proof of Theorem 1, we have

$$\mathbb{E}_{P_3}\left[e^{nf\left(\mathbf{Y},\mathbf{Y}',\hat{\mathbf{Y}},\hat{\mathbf{Y}}'\right)}\right] \leqslant n. \tag{98}$$

This completes the proof.

### G.5 PROOF OF LEMMA 1

For ease of notations, for $i \in [n]$, denote

$$\ell_{i,\pi_R} := \frac{1}{n} \mathbb{1}_{\{\hat{Y}_{\pi_R,i} \neq Y_{\pi_R,i}\}},$$

$$\ell'_{i,\pi_R} := \frac{1}{n} \mathbb{1}_{\{\hat{Y}'_{\pi_R,i} \neq Y'_{\pi_R,i}\}}.$$

Consider similar notations for the mapping $\pi$ to define $\ell_{i,\pi}$ and $\ell'_{i,\pi}$. Furthermore, denote

$$\Delta\ell := \sum_{i=1}^n (\ell_{i,\pi_R} - \ell_{i,\pi}) = \sum_{i=R+1}^n (\ell_{i,\pi_R} - \ell_{i,\pi}),$$

$$\Delta\ell' := \sum_{i=1}^n (\ell'_{i,\pi_R} - \ell'_{i,\pi}) = \sum_{i=R+1}^n (\ell'_{i,\pi_R} - \ell'_{i,\pi}).$$

It is easy to verify that $\Delta\ell = -\Delta\ell'$ and

$$|\Delta\ell| \leqslant \frac{1}{n}(n - R) = \frac{1}{2}\|\hat{p}_{\mathbf{Y}} - \hat{p}_{\mathbf{Y}'}\|_1. \tag{99}$$

With these notations,

$$\begin{aligned}
f\left(\mathbf{Y}_{\pi_R}, \mathbf{Y}'_{\pi_R}, \hat{\mathbf{Y}}_{\pi_R}, \hat{\mathbf{Y}}'_{\pi_R}\right) &= h_D\left(\sum_{i=1}^n \ell'_{i,\pi_R}, \sum_{i=1}^n \ell_{i,\pi_R}\right) - h_{\mathbf{Y},\mathbf{Y}',\hat{\mathbf{Y}},\hat{\mathbf{Y}}'}\left(\frac{1}{2}\|\hat{p}_{\mathbf{Y}} - \hat{p}_{\mathbf{Y}'}\|_1\right) \\
&\overset{(a)}{\leqslant} h_D\left(\sum_{i=1}^n \ell'_{i,\pi_R} - \Delta\ell', \sum_{i=1}^n \ell_{i,\pi_R} - \Delta\ell\right) \\
&= h_D\left(\sum_{i=1}^n \ell'_{i,\pi}, \sum_{i=1}^n \ell_{i,\pi}\right), \tag{100}
\end{aligned}$$

which completes the proof, assuming the step (a) holds.

It then remains to show the step $(a)$. To show this step, it is sufficient to prove that for every $x_1, x_2 \in [0,1]$, $\tilde{\epsilon} \in \mathbb{R}^+$, and $\epsilon \in \mathbb{R}$ such that $(x_1 + \epsilon), (x_2 - \epsilon) \in [0,1]$ and $|\epsilon| \leqslant \tilde{\epsilon}$, the below inequality holds:

$$h_D(x_1, x_2) - h_C(x_1, x_2; \tilde{\epsilon}) \leqslant h_D(x_1 + \epsilon, x_2 - \epsilon). \tag{101}$$

Without loss of generality, assume that $x_1 \leqslant x_2$. We show the above inequality for different ranges of $\epsilon$, separately.

- If $\epsilon \leqslant 0$, then since by (SZK23, Lemma 1), $h_D(x; x_2)$ is decreasing in the real-value range of $x \in [0, x_2]$ and $h_D(x_1; x)$ is increasing in the real-value range of $x \in [x_1, 1]$, we have

$$h_D(x_1, x_2) - h_D(x_1 + \epsilon, x_2 - \epsilon) \leqslant 0$$
$$\leqslant h_C(x_1, x_2; \tilde{\epsilon}),$$

where the last inequality follows using the fact that $h_C$ is non-negative.

- If $\epsilon \geqslant x_2 - x_1$, then by letting $\epsilon' = (x_2 - x_1) - \epsilon \leqslant 0$, we have

$$h_D(x_1, x_2) - h_D(x_1 + \epsilon, x_2 - \epsilon) = h_D(x_1, x_2) - h_D\big(x_2 - \epsilon', x_1 + \epsilon'\big)$$
$$\overset{(a)}{=} h_D(x_1, x_2) - h_D\big(x_1 + \epsilon', x_2 - \epsilon'\big)$$
$$\overset{(b)}{\leqslant} 0$$
$$\overset{(c)}{\leqslant} h_C(x_1, x_2; \tilde{\epsilon}),$$

where $(a)$ is deduced by the symmetry of $h_D$ and steps $(b)$ and $(c)$ are deduced similar to the case $\epsilon \leqslant 0$ above.

- If $\epsilon \in [0, (x_2 - x_1)/2]$, then we have

$$h_D(x_1, x_2) - h_D(x_1 + \epsilon, x_2 - \epsilon) = h_b(x_1 + \epsilon) + h_b(x_2 - \epsilon) - h_b(x_1) - h_b(x_2)$$
$$\leqslant h_C(x_1, x_2; \tilde{\epsilon}),$$

where the last step follows by definition of the function $h_C$, and since $\epsilon$ belongs to the below interval:

$$[0, \tilde{\epsilon}] \cap [0, (x_{1 \vee 2} - x_{1 \wedge 2})/2]. \tag{102}$$

- If $\epsilon \in [(x_2 - x_1)/2, (x_2 - x_1)]$, then by letting $\epsilon' = (x_2 - x_1) - \epsilon$, we have $\epsilon' \in [0, (x_2 - x_1)/2]$ and

$$h_D(x_1, x_2) - h_D(x_1 + \epsilon, x_2 - \epsilon) = h_b(x_1 + \epsilon') + h_b(x_2 - \epsilon') - h_b(x_1) - h_b(x_2)$$
$$\leqslant h_C(x_1, x_2; \tilde{\epsilon})$$

where the last step follows by definition of the function $h_C$, and since $\epsilon$ belongs to the below interval:

$$[0, \tilde{\epsilon}] \cap [0, (x_{1 \vee 2} - x_{1 \wedge 2})/2]. \tag{103}$$

Note that $\epsilon' \leqslant \tilde{\epsilon}$, since $\epsilon' \in [0, (x_2 - x_1)/2]$ and $\epsilon \in [(x_2 - x_1)/2, (x_2 - x_1)]$. Hence, $\epsilon' \leqslant \epsilon$, and by assumption $\epsilon \leqslant \tilde{\epsilon}$.

This completes the proof of the lemma.

