# OpenReview forum: "Generalization Guarantees for Representation Learning via Data-Dependent Gaussian Mixture Priors"
_ICLR.cc/2025/Conference — ICLR 2025 Spotlight_

### Official Review · Reviewer_9bKp · 2024-10-20

**Soundness:** 3
**Presentation:** 3
**Contribution:** 3
**Rating:** 8
**Confidence:** 2

**Summary:**

This paper provides a new generalization bound for representation learning in multi-class classification tasks. The error bound incorporates the recent notion of minimum description length (MDL), which empirically has proven to be more useful than mutual information-based bounds. In particular, the in-expectation bound and tail bound only depend on the encoder part. This has implications that in order to achieve better generalization bound, one can propose a regularizer based on the MDL. This is further studied by approximations using a Gaussian mixture whose mean and variance are data-dependent, and an update scheme for the prior is provided. Several numerical experiments verify the theory.

**Strengths:**

1. The paper is clearly written and well-organized.

2. The derived bound is tighter than the best bound with MDL.

3. This work provides a quantitative explanation that the encoder plays the role of generalization, as reflected in the bounds in Theorem 1 and 2.

4. Based on the regularization using MDL, a practical and explicit optimization scheme for the prior is provided using Gaussian mixture models.

5. The theory is verified on a few datasets.

**Weaknesses:**

See questions.

**Questions:**

1. Typo in line 22 of the abstract, "date-dependent" should be "data-dependent".

2. In line 49, the author should make it clear what "MI" stands for before writing "MI-based".

3. In line 86, the author should cite earlier references about the universal approximation with Gaussian mixture, for example, some articles in JRSSB.

4. How do you determine the number of modes in the Gaussian mixture prior?

---

> ### Author Response · Authors · 2024-11-19
> **Response to Reviewer 9bKp**
>
> We thank the reviewer for the relevant comments.
>
> 1. Thank you. Modified.
>
> 2. In the revised version, we made it clearer by writing "mutual information (MI)" in the second paragraph of the introduction.
>
> 3. We agree with the reviewer and have added [DH83] and [GBC16]. If the reviewer recommends any other related references, we would be thankful to share them with us.
>
>    [DH83] SR Dalal and WJ Hall. Approximating priors by mixtures of natural conjugate priors. Journal of the Royal Statistical Society: Series B (Methodological), 45(2):278–286, 1983.
>
>    [GBC16] Ian Goodfellow, Yoshua Bengio, and Aaron Courville. Deep learning, 2016.
>
> 4. The suitable choice of the value of the parameter $M$ depends on several factors, especially the dimension $d$ of the latent variables and the "sparsity" or number of "subpopulations" in the latent variables (which depends on the complexity of the data and the complexity of the encoder). Therefore, in general, it is not possible to know precisely an appropriate value of $M$ in advance and should be hence considered as a hyperparameter.
>
>    As future work, it would be interesting to investigate how common dimensionality reduction and unsupervised clustering techniques, such as the t-SNE dimensionality reduction and clustering using a Gaussian mixture, could be used to obtain a good "initial guess" for the choice of $M$.

---

> > ### Comment · Reviewer_9bKp · 2024-11-22
> >
> > Thank you for the response and the revision. I have no further concerns and would like to keep my rating.

---

> > > ### Author Response · Authors · 2024-11-23
> > >
> > > Thank you for your positive feedback and relevant comments.

---

### Official Review · Reviewer_w8tc · 2024-11-01

**Soundness:** 3
**Presentation:** 2
**Contribution:** 3
**Rating:** 6
**Confidence:** 3

**Summary:**

In this paper, the generalization performance of representation learning algorithms (where the models consist of an encoder and a decoder) is considered. Specifically, bounds based on the compressibility of the latent variables of the models are derived. On the basis of these bounds, regularizers using priors with a Gaussian mixture model are devised. These regularizers are numerically shown to provide improved performance compared to prior work for some tasks.

**Strengths:**

The general problem under consideration is interesting, and the twin goals of explaining generalization and finding ways to improve it are valuable. Interesting connections are drawn between, e.g., the presented bounds and classical compressibility, and between the form of the prior updates and attention. The way in which the regularizer is constructed is, as far as I am aware, new and yields promising results. The numerical experiments compare to several prior alternatives from the literature.

**Weaknesses:**

The presentation is not always clear. For instance, one of the key concerns about the validity of the data-dependent prior, as far as I understood, is to guarantee that it is symmetric with respect to the training data and a ghost sample. However, the procedure seems to only depend on the training data, and it was unclear to me whether symmetry was preserved. Noise was added at some stages to “partially” preserve symmetry, and allusions were made to prior work which allows for conditions to be only partially fulfilled (e.g. differential privacy), but the results were discussed for exact symmetry.

The relation between prior literature and the presented generalization bounds and data-dependent priors could be discussed to a greater extent.

Minor:

— “date-dependent” in abstract

— Mix of numbering and words in some lists (“(1) First” in line 218, Lines 350-356)

**Questions:**

1. Can you clarify the data-dependence of the prior, and how this relates to the assumptions under which Theorem 1 is derived?

2. For Theorem 1, the prior to be symmetric (equivalent to the exchangeable priors of Audibert (“A better variance control for PAC-Bayesian classification”, 2004) and Catoni (“PAC-Bayesian Supervised Classification: The Thermodynamics of Statistical Learning”, 2007). In their work, it is often sufficient to consider “almost exchangeable priors”, where permutations are restricted to only swap the $i$th element of the training sample with the $i$th sample in the ghost sample. Would a similar weaker requirement work for your results?

3. Line 266: “when the empirical risk is set to 0.05” What does this mean exactly? Does it mean that you train until the empirical risk reaches 0.05 or below and then stops?

4. In Theorem 1, the bound is provided in terms of a scaled Jensen-Shannon divergence between two Bernoullis. This is reminiscent of the Maurer-Langford-Seeger (MLS) bound in PAC-Bayes, where the corresponding KL divergence is used in the LHS. There are results (Foong et al, “How Tight Can PAC-Bayes be in the Small Data Regime?”, Thm. 4, 2021) indicating that the MLS bound is the tightest possible bound (in some sense) up to the log term. What is the relation between Thm. 1 in the present paper and such preceding bounds? Is it possible to instead use the binary KL in the LHS? (Such bounds also have a fast-rate behavior for zero training loss). I understand that the present paper considers a slightly different setup.

---

> ### Author Response · Authors · 2024-11-19
> **Response to Reviewer w8tc (1/2)**
>
> We sincerely thank the reviewer for the relevant comments and suggestions that helped improve our paper.
>
> **Regarding the literature:** We have added a brief discussion of how our approach compares to previous related work.
>
>
> **Questions:**
>
> 1. We thank the reviewer for bringing up this discussion. Generally, it is a common practice to design practical approaches that are "inspired" by the theoretical results by deviating slightly from the required assumptions. Furthermore, to make the proposed approach more founded, in the initial version (footnote on page 7) we briefly mentioned how the established bounds can be extended to cases where the symmetry is slightly violated using differentially private tools, e.g. ones similar to [DR18]. In fact, there are various ways to do such an extension and we believe a separate dedicated work is required to study them. However, to address the reviewer's concern, in Appendix~B of the revised versions we have detailed the explanations in the footnote and proposed two preliminary methods to extend the results to allow a slight violation of the symmetry condition (with a slight penalty). The first one is by using the differentially private prior tools (as was mentioned in the footnote on page 7 of the original manuscript) and the second one is by a new definition of a "partially symmetric prior". As a future work, it would be interesting to quantify more precisely the amount of "violation" from the symmetry of the prior in our proposed approach. Here, we would like to emphasize that in particular in the few first epochs, which have been reported in many works to be critical for the rest of the training procedure, such violation is intuitively small since the model has not yet "memorized" the training dataset, and both adding noise and passing over different batches has the effect of "forgetting" which reduces the dependence of the prior on the next batch.
>
> 2. Thanks for such a keen remark. First, we would like to elaborate a bit onto why, and how, the type of symmetry that is needed to derive bounds in terms of latent variable complexity is different from those that appear, e.g., in the mentioned exchangeable priors of Audibert. In those  works, as mentioned by the reviewer, we swap some training samples with test samples. This will result in a new training dataset and since the decoder part also depends on the training dataset, this kind of shuffling yields bounds that involve, among other terms, one that accounts for the "decoder" or "prediction" complexity. This contrasts with our work here which seeks establishing bounds on the generalization that depend on the encoder complexity only (via the latent variable complexity), not the decoder complexity. Hence, in the type of symmetry that we choose we shuffle only the assigned latent variables (under label preservation) while keeping the training and test dataset as before. This is the reason for which in our definition of the needed symmetry for our results we require $\mathbf{Q}(U^{2n}_{\pi}|Y^{2n},X^{2n},W_e)$ to be invariant under all label-preserving permutations $\pi\colon [2n]\mapsto [2n]$. Here, we swap the latent variables but keep training and test datasets unchanged. Such analyses are in general more delicate.
>
>    That being said, in this work, we consider maximal shuffling of latent variables $\mathbf{U}$ and $\mathbf{U}'$ for training and test datasets; that preserves the labeling (i.e. we shuffle two latent variables if they are related to samples that have the same label). Instead, as can be concluded naturally by the comment of the reviewer, we could only consider shuffling one latent variable in the training dataset with only one latent variable in the test dataset that has the same label as the training sample. Such an approach would give the bound established in [SZK23, Theorem 4] with order of intuitively $\sqrt{\text{MDL}(\mathbf{Q})/n}$. The maximal shuffling allows to improve the bound to order $\text{MDL}(\mathbf{Q})/n$ in some cases.

---

> > ### Author Response · Authors · 2024-11-19
> > **Response to Reviewer w8tc (2/2)**
> >
> > 3. On the value 0.05 of the empirical risk used for the simulation results: first, note that this here refers to the empirical training error, not the objective function that is minimized in practice, e.g. using SGD. In practice, the training continues until the objective function (or its gradient norm) is negligible. If the objective function is set as the empirical risk, then minimizing the former will result in a very small empirical risk. However, whenever a regularization term is added to the objective function, the final empirical risk is not necessarily small. For example, in our approach, we observed that the highest test accuracy is often achieved when there is a suitable trade-off between the empirical risk and the regularizer, resulting in the empirical risk being in the range $[0.05,0.2]$, depending on the setup.  In the revised version, we have plotted the bounds for different values of the empirical risk for better clarity.
> >
> > 4. In general, we have
> > \begin{align}
> >     h_D(x,x')=&2D_{JS}(\text{Bern}(x)\|\text{Bern}(x'))
> >     = D_{KL}\left(\text{Bern}(x)\|\text{Bern}((x+x')/2)\right) + D_{KL}\left(\text{Bern}(x')\|\text{Bern}((x+x')/2)\right).
> > \end{align}
> > Hence, there is indeed a relation between our bounds and binary KL-divergence terms. However, unfortunately, we are not aware of any approach to obtain (at least trivially) a result in terms of latent variable complexity using the binary KL function, as is often used in the literature (e.g. by Foong et al). There are two main difficulties. The first is due to the fact that such approaches are mainly in terms of "model complexity". Since the dimension of the model is generally large, it is not computationally efficient to have regularizers in terms of the model. Moreover, in practice, we are given a single sample of the model, and estimating a suitable posterior distribution of the model is more challenging. The second problem is that such approaches also seem to depend on the decoder part, an aspect which does not seem aligned with our purpose in this work of establishing bounds that only depend on the encoder complexity.

---

> > > ### Comment · Reviewer_w8tc · 2024-11-21
> > >
> > > Thank you for your response.
> > >
> > > 1. That's fine, but my impression is that this was not made very clear in the paper itself. If the regularizer that you use is only inspired by the bound, but not actually obtainable through a valid prior (and hence not a regularizer that can directly pop out from the bound except via approximations), this should be clearly stated to avoid misleading readers. I apologize if this was already made clear in the initial submission, but if it was, I seem to have missed it.

---

> > > > ### Author Response · Authors · 2024-11-21
> > > >
> > > > Thank you for your feedback. Actually, your point was a good one.
> > > >
> > > > However, please note that:
> > > >
> > > > - It was stated in the orignal version (see the footnote of page 7 of the initial submission) that the prior does not need to satisfy the very exact symmetry condition of Definition 1 (i.e., that exact symmetry requirement can be relaxed partially and still the bound of Theorem 1 would still hold (at the expense of some small penalty); and, this is why the prior was considered to be a valid one in that version.
> > > >
> > > > - Now, in the revision, not only we make the aforementioned footnote more precise; but also we provide the details of the extension of the result of Theorems 1 and 2 to the practical setting of partial symmetry (see Propositions 1 and 2 in Appendix B).
> > > >
> > > > In summary, we hope that with these clarifications the validity of the prior as constructed in the paper is now clearer to you. Please let us know if this is now fine for you; and whether you have any additional comments, in which case we would be happy to address.

---

### Official Review · Reviewer_ce5p · 2024-11-02

**Soundness:** 3
**Presentation:** 2
**Contribution:** 3
**Rating:** 8
**Confidence:** 3

**Summary:**

The paper studies the generalization error of representation learning, in which we have the encoder-decoder model. The authors establish a new generalization bound based on the minimum description length (MDL) of a symmetric prior and the induced representation distribution of the encoder. The paper explains how to choose the prior using a mixture of Gaussians, and connects the generalization bound to the regularizer of the optimization problem, in which the prior and the representation distribution are learned jointly. The authors examine their method in simple datasets and model architectures, and in both lossy and lossless scenarios.

**Strengths:**

The paper is well-written and the motivation is clear. The authors improved the generalization bound of the encoder-decoder representation learning from $\sqrt{\text{MDL}(Q)/n}$ to $\text{MDL}(Q)/n$. The proposed symmetric prior using mixtures of Gaussians is relatively practical and easy to implement. The idea of adding the generalization bound as a regularizer and learning the prior and the induced distribution of the encoder together is interesting.

**Weaknesses:**

While the authors have improved on the previous generalization bound, the technical work and the idea of regularization are heavily based on [1]. However, I think the mixture of Gussians is a nice addition.

The lossy generalization bound in Section 3.3 seems a bit incomplete to me. While the authors explain what lossy means, I could not follow how it results in Eq 14 and 15. I also checked the appendix, but could not find anything related to this section. I would appreciate it if the authors could elaborate on this section more, or add a theorem and the proof either in the main text or the appendix.

I also think there are some parts that the author can improve the writing or the intuition of their work (see questions).

[1] Minimum Description Length and Generalization Guarantees for Representation Learning, Sefidgaran et al. 2023

**Questions:**

1. In section 3.3, the bound in Eq 14 seems to be similar to [1] with $\sqrt{\text{MDL}(Q)/n}$ rather than $\text{MDL}(Q)/n$. Do the authors have any intuition/proof on why this change happens in the lossy setting?

2. Is there any intuition behind $h_C$ in Eq 6?

3. In section 4.1, M denotes the number of mixtures. Do the authors suggest any systematic way of choosing M? Or is it a hyperparameter that needs to be tuned?

Some typos also can be addressed:
- Line 342, an addition | in KL
- Eq. 17, the index of X should be batch size and not beta

---

> ### Author Response · Authors · 2024-11-19
> **Response to Reviewer ce5p(1/2)**
>
> We thank the reviewer for the careful reading of the paper and the comments, which are relevant.
>
> **Regarding the difference with [1]:** Indeed, as mentioned by the reviewer, the design of prior as a mixture of Gaussians is a significant addition, comparatively. However, we kindly attract the attention of the reviewer on that, in addition to this one, there are other important differences with the approach of [1], especially in the proof techniques of our main results which are required to get our tighter generalization bounds, with a better dependence on the number of training samples $n$ (namely, $\text{MDL}(\mathbf{Q})/n$ instead of the $\sqrt{\text{MDL}(\mathbf{Q})/n}$ of [1]). For instance, as we now elaborate after Theorem 1 and our response to Reviewer C35E (see the part "Regarding the proof techniques" of the rebuttal to that reviewer), unlike [1] we do not find another dataset having the same labels as the training dataset, as this introduces the penalty of order $\sqrt{\frac{C}{n}}$, and we analyze directly the terms (such as computing moment-generating functions) for $S$ and $S'$, which have potentially unequal numbers of samples per label. Secondly, for each label $c\in[C]$, we consider all samples in $S$ and $S'$ that have the label $c$. Denoting this set as $\mathcal{B}_c$, our proof then relies on the complete reshuffling of the latent variables corresponding to the samples in the set $\mathcal{B}_c$. Analyzing the complete reshuffling of the latent variables, especially taking into account the unequal number of samples per label, makes the analysis more involved (for example applying Donsker-Varadhan on $nf$ as defined in the proof of Theorem 1, rather than $nh_D$, and bounding the moment-generating function of $nf$ rather than $nh_D$). But, it is precisely these two steps that allow us to obtain bounds that behave as $\text{MDL}(\mathbf{Q})/n$ in some cases.

---

> ### Author Response · Authors · 2024-11-19
> **Response to Reviewer ce5p(2/2)**
>
> **Questions**
>
> 1. **On the lossy compressibility and Equations (14) and (15):** First, note that, by assumption, we have $\mathbb{E}\left[\text{gen}(S,W)-\text{gen}(S,\hat{W})\right] \leq \epsilon$, or equivalently $\mathbb{E}\left[\text{gen}(S,W)\right] \leq \mathbb{E}\left[\text{gen}(S,\hat{W})\right]+\epsilon$. Therefore, it is sufficient to upper bound $\mathbb{E}\left[\text{gen}(S,\hat{W})\right]$ for the "quantized model" $\hat{W}=(\hat{W}_e,W_d)$. This is easily done by invoking [SZK23, Theorem~4]; and it yields (14) and (15).
>
>    Note that this simple trick has both practical and conceptual implications. Conceptually, it measures "lossy compressibility". To give an example, let's consider source compression for simplicity. While discrete random variables with a finite support set can be expressed (or "stored") using a finite number of bits, a "continuous" random variable requires an infinite number of bits for lossless description. However, if we allow the random variable to be recovered with some precision or distortion, we can express it using a finite number of bits, which is characterized by the rate-distortion function in the source coding literature. More technically, in our setup, while $\text{MDL}(\mathbf{Q})$ considered for $W$ may be infinite for deterministic encoders with continuous output support, $\text{MDL}(\mathbf{Q})$ considered for $\hat{W}$ is finite.
>
>    As already discussed in Section 3.3, we have stated the lossy compressibility result by invoking [SZK23, Theorem 4] mainly for reasons of simplicity (and this has yielded a generalization bound of the order of $\sqrt{\text{MDL}(\mathbf{Q})/n}$). However, taking into account the suggestion of the reviewer, we have added in the revised version (see Appendix~A) a lossy compressibility result as well as a detailed proof of it. Observe that the new result is expressed in terms of the function $h_D$  and has a better dependence on the number of training samples $n$, i.e.,  $\text{MDL}(\mathbf{Q})/n$ instead of the $\sqrt{\text{MDL}(\mathbf{Q})/n}$.
>
> 2. **On the intuition about the function $h_C$:** First, let us give an intuition about $h_D(x_1,x_2)$. Fix some large integer $m$. Suppose that we are given $2m$ balls where $[m(x_1+x_2)]$ of them are red and the rest of them are blue -- $[\cdot]$ denotes the integer part. We take $m$ balls randomly and without replacement from these $2m$ balls. Let $P_m$ be the probability that $[m x_1]$ of the picked balls are red. Then, $h_D(x_1,x_2) = - \lim_{m\to \infty} \frac{1}{m} \log(P_m)$. In other words, $h_D(x_1,x_2)$ is the "asymptotic error exponent" of the event of having $[m x_1]$ red ball. To put this in the context of our work, the red balls can be thought of as samples with a loss value of one. Thus, in a simple scenario, if we have a superset containing samples from the training and test datasets that have a total error of $n(x_1+x_2)$, then $h_D(x_1,x_2)$ is the asymptotic error exponent of the probability that the empirical risk is equal to $x_1$, if we choose $m$ samples randomly from the superset as the training dataset.
>
>    Now, regarding the question of the reviewer; assuming $x_1 \leq x_2$, we can re-write the $h_C$ function as  $$h\_C(x\_1,x\_2,\epsilon) = \max\_{\epsilon'}\left\\{ h\_D(x\_1,x\_2)-h\_D(x\_1+\epsilon',x\_2-\epsilon')\right\\},$$
>
>    where $\epsilon' \in [0,\min(\epsilon,(x_1-x_2)/2)]$. For a given $\epsilon'$, the term $ h_D(x_1+\epsilon',x_2-\epsilon')$ captures the asymptotic error exponent of picking $m(x_1+\epsilon')$ red balls in the above scenario (from $2m$ balls with total $m(x_1+x_2)$ red balls). Hence, $h_C(x_1,x_2,\epsilon)$ can be seen as the maximum change of the asymptotic error exponent of the probability of picking $m(x_1+\epsilon')$ red balls (w.r.t. prob. of picking $mx_1$ red balls) in the range $\epsilon' \in [0,\min(\epsilon,(x_1-x_2)/2)]$. It is worth mentioning that whenever $x_2\leq 1/2$, then the maximum is achieved for  $\epsilon'=\min(\epsilon,(x_1-x_2)/2)$.
>
> 3. **On the selection of the value of parameter $M$:** This depends on several factors, especially the dimension $d$ of the latent variables and the "sparsity" or number of "subpopulations" in the latent variables (which depends on the complexity of the data and the complexity of the encoder). Therefore, in general, it is not possible to know precisely an appropriate value of $M$ in advance and should be hence considered as a hyperparameter.
>
>    As future work, it would be interesting to investigate how common dimensionality reduction and unsupervised clustering techniques, such as the t-SNE dimensionality reduction and clustering using a Gaussian mixture, could be used to obtain a good "initial guess" for the choice of $M$.

---

### Official Review · Reviewer_6GSR · 2024-11-04

**Soundness:** 3
**Presentation:** 2
**Contribution:** 2
**Rating:** 6
**Confidence:** 2

**Summary:**

This paper provides generalization guarantees for representation learning algorithms by deriving in-expectation and tail bounds on generalization error. The authors develop these bounds based on the MDL principle, using a data-dependent Gaussian mixture prior as a regularizer. By establishing bounds related to the relative entropy between representations from training and test datasets, the method aims to leverage the simplicity and structure of the encoder for improved generalization.

**Strengths:**

1. The paper provides non-vacuous generalization bounds for representation learning.

2. The proposed regularizer has been validated in many image datasets.

3. Theoretical results have been rigorously presented and supported.

**Weaknesses:**

1. The Gaussian mixture prior introduces additional computational overhead, especially during training, which may make the approach less practical for very large datasets.

2. Although the emergence of a weighted attention mechanism is highlighted, the paper could benefit from a more detailed analysis of this component and its role in generalization.

3. The empirical validation is limited to standard classification datasets. Testing on real-world tasks beyond classification (e.g., transfer learning, semi-supervised learning) would strengthen the claims of generalization benefits.

**Questions:**

1. What might be the potential limitations of the current method? What could be the future work?

2. Are there any constraints on the applicability in real-world scenarios introduced by relying on a Gaussian mixture prior?

---

> ### Author Response · Authors · 2024-11-19
> **Response to Reviewer 6GSR**
>
> We thank the reviewer for the relevant comments and suggestions.
>
> **Weaknesses:**
>
> 1. First, we emphasize that in our approach, the regularizer is used only during the training phase. Thus, there is no overhead in the inference phase. Moreover, similar to VIB, our regularizer depends on the dimension of the latent variable, rather than on the dimension of the model or the input data, which are often much larger. Moreover, our approach is rather easy to implement, as acknowledged for example by Reviewer ce5p.
>
>    However, we agree with the reviewer that, similar to many ones that introduce/use regularizers, the gains of the approach come at the expense of a slight addition in the computational cost. Possible approaches towards reducing further that overhead could include: (i) using the regularizer only in the first $K$ epochs (which, usually, are the most critical ones for convergence) and (ii) first projecting the latent space onto a lower dimensional one and then applying the regularizer in that lower-dimensional space. We now elaborate more on this in the newly added "Future Directions" section in Appendix D.
>
> 2. Regarding the connection with the attention mechanism, we agree with the reviewer that further investigations along this direction are very interesting. In particular, in combination with point 1 above, we expect that when our approach is employed in the models that use the self-attention mechanism some of the computations performed therein could be used to update the GM prior and thus partially reduce the computational overhead.
>
> 3. Please note that our theoretical results of Theorem 1 and Theorem 2 are established for the classification setup and naturally the experiments are performed only in this setup (but on various datasets, as already noted by the reviewer). In fact, we agree that it would be interesting to extend both the theoretical and experimental results of this work to use in transfer learning and semi-supervised learning, which is beyond the scope of the current work. For example, in transfer learning, can the learned priors be used for the transfer learning task? Or in semi-supervised learning, can unlabeled data be used to find the GM prior more accurately? In the revised version, we have discussed such aspects in the newly added "Future Directions" section.
>
>
> **Questions:**
>
> 1. We discussed some limitations (e.g. computational complexity and being limited to classification) and future directions (e.g. computational efficiency, considering our approach in the self-attention layer, and extending to transfer learning and semi-supervised learning setups) in the revised version.
>
> 2. As discussed in the paper, the Gaussian mixture is chosen for two reasons: (i) the Gaussian mixture distribution is known to possibly approximate any arbitrary distribution well enough, provided the number of mixture components is sufficiently large [NND+22]; and (ii) given distributions $\\{p_{i}\\}\_{i\in[N]}$, the distribution $q$ that minimizes $\sum\_{i\in[N]} D\_{KL}(p\_i\||q)$ is $q=\frac{1}{N}\sum\_{i\in[N]}p\_i$. Thus, if all $p\_{i}$ distributions are Gaussian, the minimizer is a Gaussian mixture.
>
>    In many of the variational encoders, the posterior is indeed Gaussian, and thus it is optimal to consider GM. Moreover, in deterministic encoders, by using the lossy approach, we can consider the posterior to be a Gaussian with the mean being equal to the deterministic encoder output and the covariance being equal to $\varepsilon\mathrm{I}\_d$, where $\varepsilon$ is the lossy compressibility hyperparameter. Thus, in most situations, the choice of GM is a natural one.

---

### Official Review · Reviewer_c35E · 2024-11-05

**Soundness:** 3
**Presentation:** 3
**Contribution:** 4
**Rating:** 8
**Confidence:** 3

**Summary:**

This paper studies the setting of representation learning, where the task is to jointly learn a good representation and a good predictor. PAC-like generalization bounds are provided, which crucially depend on the complexity of the latent representation. Based on these theoretical insights, the authors then suggest a new regularizer term, which shows promising results in some applications.

**Strengths:**

- Novel generalization bounds for representation learning: The paper derives generalization guarantees for representation learning. To my knowledge, this setting has not been studied in depth previously. I found the remarks about these bounds interesting, particularly the fact that they do not depend on the encoder component.
- Novel regularization term based on the theoretical analysis: As is often the case with generalization bounds, they can directly inspire regularization terms or other modifications to the learning pipeline. The authors propose a new regularization term for representation learning. In experiments with image classification datasets, the introduced method is shown to compete favorably with other recently introduced methods for this task.

**Weaknesses:**

In general, I found the presentation of some results in the paper to be a bit lacking, while the discussion of related work seems somewhat insufficient. The authors discuss prior work in the introduction, but the discussion is high-level and mainly directed at 'experts' in the field. An additional (perhaps small) section covering related work would have served this paper well. Additionally, the main results of the paper (Theorem 1 and Theorem 2) are presented without much (if any) discussion on the techniques used in their proofs. Another example is in line 305, where the authors mention and emphasize the 'geometrical compression phenomenon' but do not describe it further. In various places, the manuscript would benefit from additional polishing.

**Questions:**

I have some concrete questions:
- What is the "joint" (line 1188, Appendix C.3) learning procedure that you followed in the experiments? Can you provide more details on the loss used, etc?
- How do the proof techniques compare to the ones used by SZK23?

---

> ### Author Response · Authors · 2024-11-19
> **Response to Reviewer c35E**
>
> We thank the reviewer for the interest in the work and relevant comments.
>
> **Regarding the related works:** We have added a brief discussion of how our approach compares to previous related work.
>
> **Regarding the proof techniques:** As suggested by the reviewer, we now add in the revised version, right after Theorem 1, a sketch of its proof with emphasis on the main elements of that proof. In what follows, we focus in particular on the reviewer's question about the difference between our proof techniques and those in [SZK23].
>
> While, at a high level, the proof techniques used in our work and those of [SZK23] can be viewed as being both based on the idea of "shuffling" of the latent variables associated with the training and test samples that have the same label, ours are more involved and the analysis is deeper, comparatively. This explains why the resulting bounds are tighter than those of [SZK23]. In order to show this, for convenience, we briefly recall the main ideas of the proof technique of [SZK23]. In the approach of [SZK23], given the datasets $S$ and $S'$, the authors (implicitly) find another dataset $\tilde{S}$ such that: (i) the vectors of labels $\tilde{\mathbf{Y}}$ (associated to $\tilde{S}$) and $\mathbf{Y}$ (associated to $S$) are equal up to some suitable re-ordering, (ii) the found dataset $\tilde{S}$ and the ghost dataset $S'$ differ in no more than $\sqrt{\frac{C}{n}}$ samples (since $\mathbf{Y}$ and $\mathbf{Y}'$ are drawn independently, with some suitable reordering of $\mathbf{Y}'$ they differ in about  $\sqrt{\frac{C}{n}}$ indices). This first proof step introduces a penalty of order $\sqrt{\frac{C}{n}}$ into the bound. Thus, with such an approach, obtaining bounds with better dependence on $n$ than $1/\sqrt{n}$ seems elusive. Next, assuming that $\tilde{Y}$ (after suitable reordering) is equal to $Y$, the rest of the proof uses arguments similar to those of the conditional mutual information (CMI) approach; and, for instance, it relies on shuffling the latent variables corresponding to the $i$th samples of the training and test datasets. This second proof step yields bounds of the order of $1/\sqrt{n}$.
>
>
> Based on the above, it appears clearer that in order to obtain new bounds with better dependence on $n$ than those of [SZK23] one needs different, 'finer', proof techniques in both aforementioned steps. This is precisely what we do.  First, we do not find another dataset having the same labels as the training dataset, as this introduces the penalty of order $\sqrt{\frac{C}{n}}$, and we analyze directly the terms (such as computing moment-generating functions) for $S$ and $S'$, which have potentially unequal numbers of samples per label. Secondly, for each label $c\in[C]$, we consider all samples in $S$ and $S'$ that have the label $c$. Denote this set by $\mathcal{B}_c$. Our proof then relies on the complete reshuffling of the latent variables corresponding to the samples in the set $\mathcal{B}_c$. Analyzing the complete reshuffling of the latent variables, especially taking into account the unequal number of samples per label, makes the analysis more involved (for example applying Donsker-Varadhan on $nf$ as defined in the proof of Theorem 1, rather than $nh_D$, and bounding the moment-generating function of $nf$ rather than $nh_D$). But, it is precisely these two steps that allow us to obtain bounds that behave as $\text{MDL}(\mathbf{Q})/n$ in some cases.
>
> **Regarding the geometrical compression:** In the revised version, we have added a short explanation in the introduction section of what is meant by "Geometric compression", a term already used in related works such as [AG19, GK19]. More precisely, "Geometric compression occurs when latent vectors are designed so as to concentrate around a limited number of representatives which form centroid vectors of associated clusters [AG19, GK19]. In such settings, inputs can be mapped to the centroids of the clusters that are closest to their associated latent vectors (i.e., lossy compression); and this yields non-vacuous bounds at the expense of only a small (distortion) penalty. The benefit of this lossy compression approach can be appreciated when opposed to classic MI-based bounds [VPV18, KDJH23] which are known to be vacuous when the latent vectors are deterministic functions of the inputs."
>
>
> **Regarding the experiments:** By joint training, we mean joint training of the encoder and decoder parts of the model. This was stated as such in order to emphasize the dependence of both encoder and decoder parts on the dataset. In the revised version, we removed this to avoid any confusion.
>
> The considered loss function is the cross entropy function, as often considered in classification tasks. This is now stated more clearly in the documents. Also, further details about the experiments are added in Appendix E.

---

> > ### Comment · Reviewer_c35E · 2024-11-20
> > **Reply**
> >
> > Thank you for your detailed answer. It is much appreciated. I believe that the paper has been improved. Some of the technical contributions are clearer now, so I decided to increase my score. I keep my low confidence score, however, to indicate that I am not very familiar with this area of research, so my opinion should be weighted accordingly.

---

> > > ### Comment · Reviewer_c35E · 2024-11-20
> > > **Minor**
> > >
> > > An additional minor comment: the font of the paper seems to be different than the official one. Please fix it.

---

> > > ### Author Response · Authors · 2024-11-21
> > >
> > > Thank you so much for the positive feedback, and again for the comments which were relevant.

---

> ### Author Response · Authors · 2024-11-21
>
> We have now changed it. Thank you for bringing our attention on that.

---

### Author Response · Authors · 2024-11-19
**Regarding the revised version**

We thank all reviewers for their valuable comments and are pleased to see that overall the reviewers found our work and approach interesting and promising. We have addressed all of the reviewers' comments separately and kindly ask them to let us know if any further clarification is needed.

We have also uploaded a revised version of the paper based on the reviewers' comments. To summarize, the main changes, which are highlighted in blue in the paper, include

- As suggested by Reviewer c35E and Reviewer w8tc, we now elaborate more on the relation with some related prior-art works.

- As suggested by Reviewer c35E, we have added, in the text, right after Theorem 1, some intuitions, insights, and elements of its proof, which we hope will help the reader better understand the result.

- We have enriched the numerical evaluation of our result of Theorem 1  reported in Figure 3 by providing figures for various values of the empirical risk (instead of one value, 0.05, in the original version) - This was also suggested by Reviewer w8tc. Interestingly, the new plot shows that smaller empirical risks result in a greater improvement of our results with respect to the previous bounds.

- We now present, in Appendix A as suggested by Reviewer CE5P, the lossy version of Theorem 1.

- In Appendix B, we have detailed the approach (previously briefly outlined in the footnote on page 7) to extend our results to the cases where the priors are "partially" symmetric, as requested by Reviewer w8tc.

- In Appendix D, we discuss the limitations of the current work and how the current work opens up several interesting future research directions, as suggested by Reviewer 6GSR. In this section, we have also discussed some potentially efficient practical approaches to choosing the number of components $M$, as suggested by Reviewer 9BKP and Reviewer CE5P, using dimensionality reduction and clustering approaches.

---

### Meta-Review · Area_Chair_Qw7r · 2024-12-20

**Metareview:**

a) Scientific Claims and Findings

This paper establishes novel generalization bounds for representation learning algorithms by leveraging the Minimum Description Length (MDL) principle and a data-dependent Gaussian mixture prior. Specifically, it provides in-expectation and tail bounds that depend on the relative entropy between the learned representations of training and test datasets and a data-dependent prior. Inspired by these bounds, the authors propose a regularizer that incorporates a learned Gaussian mixture prior, showing promising results in outperforming existing methods such as Variational Information Bottleneck (VIB) and Category-Dependent VIB (CDVIB). The proposed method also demonstrates connections to weighted attention mechanisms.

(b) Strengths

Rigorous derivation of generalization bounds that improve upon existing approaches in terms of encoder complexity.
Novel and practical use of data-dependent Gaussian mixture priors to devise a regularization strategy.
Comprehensive theoretical analysis coupled with numerical experiments showcasing superior performance over competing methods.
The paper addresses critical concerns raised by reviewers, enhancing its clarity and robustness.

(c) Weaknesses

The computational cost introduced by the Gaussian mixture prior during training may hinder scalability to large datasets.
Empirical validation is limited to standard classification tasks, leaving the potential for broader applications unexplored.
Some results, such as the geometrical compression phenomenon and the role of attention mechanisms, require further elaboration.

(d) Decision Rationale

I recommend acceptance of this paper. It makes a significant theoretical contribution by tightening generalization bounds and providing practical insights for representation learning. The experimental results substantiate the theoretical claims, demonstrating utility and innovation. While some aspects could benefit from additional experiments or clarity, the paper’s contributions are sufficiently impactful to merit inclusion in the conference.

**Additional Comments On Reviewer Discussion:**

During the rebuttal phase, the authors effectively addressed most concerns raised by the reviewers:

Reviewer c35E: Highlighted a lack of related work discussion and clarity in proofs. The authors improved the discussion on related work and provided insights into the proofs, leading to an increased score.

Reviewer 6GSR: Critiqued the computational overhead and limited scope of experiments. The authors clarified that the regularizer’s computational cost is manageable and proposed extensions in the "Future Directions" section. The reviewer appreciated the detailed rebuttal and increased their score.

Reviewer ce5p: Questioned the lossy generalization bounds and intuition behind specific terms. The authors added detailed explanations and proofs, particularly regarding lossy compressibility, which satisfied the reviewer and resulted in an increased score.

Reviewer w8tc: Requested clarification on the data-dependent prior’s symmetry. The authors expanded the discussion in the appendix and proposed methods for partial symmetry, which addressed the reviewer’s concerns.

Reviewer 9bKp: Suggested improvements in discussing the Gaussian mixture's design. The authors added relevant discussions, which the reviewer acknowledged positively.

Overall, the authors demonstrated responsiveness and improved the paper through the review process, contributing to my decision to recommend acceptance.

---

### Decision · Program_Chairs · 2025-01-22

Accept (Spotlight)